# Byzantine-tolerant federated Gaussian process regression for streaming data

**Xu Zhang**
Pennsylvania State University
xxz313@psu.edu

**Zhenyuan Yuan**
Pennsylvania State University
zqy5086@psu.edu

**Minghui Zhu**
Pennsylvania State University
muz16@psu.edu

## Abstract

In this paper, we consider Byzantine-tolerant federated learning for streaming data using Gaussian process regression (GPR). In particular, a cloud and a group of agents aim to collaboratively learn a latent function where some agents are subject to Byzantine attacks. We develop a Byzantine-tolerant federated GPR algorithm, which includes three modules: agent-based local GPR, cloud-based aggregated GPR and agent-based fused GPR. We derive the upper bounds on the prediction error between the mean from the cloud-based aggregated GPR and the target function provided that Byzantine agents are less than one quarter of all the agents. We also characterize the lower and upper bounds of the predictive variance. Experiments on a synthetic dataset and two real-world datasets are conducted to evaluate the proposed algorithm.

The Internet of Things (IoT) produces large volumes of spatially distributed data. A typical practice is to store data in a centralized entity, e.g., a cloud, and train machine learning models there. However, it poses challenges in communication overhead and computation efficiency. In addition, data storage on the centralized entity raises privacy concerns. Federated learning provides a promising paradigm to address the two challenges [1], [2], [3], [4]. First, only a finite number of model parameters are shared and the size of these parameters is much smaller than that of raw data; second, data samples are kept by data owners and local models are trained without sharing raw data. However, these data owners are vulnerable to Byzantine attacks, which force them to behave arbitrarily and send any message to the cloud. The cloud cannot access each agent's local private data and training processes, and then cannot verify the correctness of local updates. Byzantine attacks incur significant degradation of learning performance if they are not explicitly taken in account.

*Related work.* There have been many recent works to secure federated learning against Byzantine attacks [5], [6], [7], [8], [9]. Paper [5] proposes *Krum* as the aggregation rule where the cloud selects a gradient such that the overall distance between this gradient to a fixed number of its closest gradients is minimal. Paper [6] develops an algorithm which is based on integrated stochastic quantization, geometric median based outlier detection and secure model aggregation. Paper [7] analyzes two robust distributed gradient descent algorithms based on median and trimmed mean operations, respectively. Paper [8] proposes a high-dimensional robust mean estimation algorithm at the cloud to combat the adversary. In paper [9], the cloud partitions all the received local gradients into batches and computes the mean of each batch. After that, the cloud computes the geometric median of the batch means. The above set of papers can guarantee that the estimation error of the parameter aggregated by the cloud is upper bounded and determined by the number of Byzantine agents. However, the existing works only consider deep neural networks (DNN) as the learning model and are limited to static data and off-line learning.

On-line learning is demanded to process data that arrives sequentially across time [10], [11], [12]. Gaussian process regression (GPR) is a solution candidate to solve the problem [13], [14], [15], [16]. First of all, GPR is a non-parametric statistical learning model. With proper choice of prior covariance function and mild assumptions of the target function, GPR is able to approximate any

36th Conference on Neural Information Processing Systems (NeurIPS 2022).

continuous function [17]. Second, GPR can be implemented recursively, reducing computational complexity and memory [18]. Third, GPR is able to quantify learning uncertainties since it predicts the target function using a posterior probability distribution [19]. However, Byzantine-tolerant GPR has not been studied yet.

*Contributions.* This paper considers Byzantine-tolerant federated learning for streaming data using GPR where a cloud and a group of agents aim to collaboratively learn a latent function despite some agents are subject to Byzantine attacks. Major contributions are listed as follows:

1) We develop a Byzantine-tolerant federated GPR algorithm, which includes three modules: agent-based local GPR, cloud-based aggregated GPR and agent-based fused GPR. Specifically, the agent-based local GPR sends potentially compromised local predictions to the cloud, and the cloud-based aggregated GPR computes a global model by a Byzantine-tolerant product of experts (PoE) aggregation rule. Then the cloud broadcasts the current global model to all the agents. Agent-based fused GPR returns local predictions by fusing the models from the cloud-based GPR with those from the agent-based local GPR.

2) The predictive mean of the cloud-based aggregated GPR is theoretically guaranteed to be restricted within a neighborhood of the target function provided that Byzantine agents are less than one quarter of all the agents. The prediction error is positively proportional to the number of Byzantine agents and that of the agents with extreme values. Predictive variance is quantified by lower and upper bounds, which are identical to that of the attack-free case.

Experiments on a synthetic dataset and two real-world datasets are conducted to demonstrate that the Byzantine-tolerant GPR algorithm is resilient to Byzantine attacks. To the best knowledge of authors, this is the first paper to study Byzantine-tolerant federated learning over streaming data using GPR.

*Notations and notions.* Throughout the paper, we use lower-case letters, e.g, $a$, to denote scalars, bold letters, e.g., $\boldsymbol{a}$, to denote vectors; upper-case letters, e.g., $A$, to denote matrices, calligraphic letters, e.g., $\mathcal{A}$, to denote sets, and bold calligraphic letters, e.g., $\boldsymbol{\mathcal{A}}$, to denote spaces. Denote $I_n \in \mathbb{R}^{n \times n}$ the $n$-by-$n$ identity matrix. We denote by $\mathbb{R}$ the set of real numbers. The set of non-negative real numbers is denoted by $\mathbb{R}_+$, and the set of positive real numbers is denoted by $\mathbb{R}_{++}$. The cardinality of set $\mathcal{A}$ is denoted by $|\mathcal{A}|$. Define the distance metric $D(\boldsymbol{z}, \boldsymbol{z}') \triangleq \|\boldsymbol{z} - \boldsymbol{z}'\|$, and the point to set distance as $D(\boldsymbol{z}, \mathcal{Z}) \triangleq \inf_{\boldsymbol{z}' \in \mathcal{Z}} D(\boldsymbol{z}, \boldsymbol{z}')$. Define $\text{proj}(\boldsymbol{z}, \mathcal{Z}) \triangleq \{\boldsymbol{z}' \in \boldsymbol{\mathcal{Z}} | D(\boldsymbol{z}, \boldsymbol{z}') = D(\boldsymbol{z}, \mathcal{Z})\}$ the projection set of point $\boldsymbol{z}$ onto set $\mathcal{Z}$. Denote the supremum of a function $\eta : \boldsymbol{\mathcal{Z}} \to \mathbb{R}$ as $\|\eta\|_\infty \triangleq \sup_{\boldsymbol{z} \in \boldsymbol{\mathcal{Z}}} |\eta(\boldsymbol{z})|$. For a real number $a$, we denote the floor function by $\lfloor a \rfloor$. We denote by $\mathcal{O}(g(t))$ the limiting behavior of some function $f(t)$ if $\lim_{t \to \infty} \frac{f(t)}{g(t)} = a$ for some constant $a > 0$.

We give formal definitions of sub-Gaussian random variables with parameter $\sigma$ and Gaussian process.

**Definition 1** *[20] A random variable $X$ with mean $\mu \triangleq \mathbb{E}[X]$ is sub-Gaussian (denoted by $X \sim subG(\sigma^2)$) if there exists a positive number $\sigma$ such that*

$$\mathbb{E}\left[\exp(\lambda(X - \mu))\right] \leq \exp\left(\frac{\sigma^2 \lambda^2}{2}\right), \quad \forall \lambda \in \mathbb{R}. \tag{1}$$

**Definition 2** *[13] A Gaussian process is a collection of random variables, any finite number of which have a joint Gaussian distribution.*

## 1 Preliminary

In this section, we provide basic knowledge of GPR. Let $\eta : \boldsymbol{\mathcal{Z}} \to \boldsymbol{\mathcal{Y}}$ be the target function, where $\boldsymbol{\mathcal{Z}} \subseteq \mathbb{R}^{n_z}$ and $\boldsymbol{\mathcal{Y}} \subseteq \mathbb{R}$. Given input $\boldsymbol{z}(t) \in \boldsymbol{\mathcal{Z}}$ at time $t$, the corresponding output is given by

$$y(t) = \eta(\boldsymbol{z}(t)) + e(t), \quad e(t) \sim \mathcal{N}(0, \sigma_e^2), \tag{2}$$

where $e(t)$ is the Gaussian measurement noise and the latent function $\eta$ is sampled from a Gaussian process prior. Let training data be in the form $\mathcal{D} \triangleq (\mathcal{Z}, \boldsymbol{y})$, where $\mathcal{Z} \triangleq \{\boldsymbol{z}(1), \ldots, \boldsymbol{z}(n_s)\}$ is the set of input data and $\boldsymbol{y} \triangleq [y(1), \ldots, y(n_s)]^{\mathrm{T}}$ is the column vector aggregating the outputs. GPR aims to infer the values of the latent function $\eta$ over a set of test data points $\mathcal{Z}_* \subset \boldsymbol{\mathcal{Z}}$ using $\mathcal{D}$.

Define kernel function $k : \mathbb{R}^{n_z} \times \mathbb{R}^{n_z} \to \mathbb{R}$ that is symmetric and positive semi-definite. Let $\eta(\mathcal{Z})$ return a column vector such that the $i$-th entry of $\eta(\mathcal{Z})$ equals $\eta(z(i))$. Assume that the latent function $\eta$ is sampled from a GP prior specified by mean function $\mu$ and kernel function $k$. Then the training outputs $y$ and the test outputs $\eta(\mathcal{Z}_*)$ are jointly Gaussian distributed as:

$$\begin{bmatrix} y \\ \eta(\mathcal{Z}_*) \end{bmatrix} \sim \mathcal{N}(\begin{bmatrix} \mu(\mathcal{Z}) \\ \mu(\mathcal{Z}_*) \end{bmatrix}, \begin{bmatrix} k(\mathcal{Z}, \mathcal{Z}) + \sigma_e^2 I_{n_s} & k(\mathcal{Z}, \mathcal{Z}_*) \\ k(\mathcal{Z}_*, \mathcal{Z}) & k(\mathcal{Z}_*, \mathcal{Z}_*) \end{bmatrix}),$$

where $\mu(\mathcal{Z})$ and $\mu(\mathcal{Z}_*)$ return a vector consisting of $\mu(z(i))$ and $\mu(z_*(j))$; $k(\mathcal{Z}, \mathcal{Z}_*)$ returns a matrix such that the entry at the $i$th row and the $j$th column is $k(z(i), z_*(j))$, and $k(\mathcal{Z}, \mathcal{Z})$, $k(\mathcal{Z}_*, \mathcal{Z})$ and $k(\mathcal{Z}_*, \mathcal{Z}_*)$ are defined in an analogous way.

Utilizing identities of joint Gaussian distribution (see page 200 in [13]), GPR makes predictions of $\eta$ on $\mathcal{Z}_*$ based on dataset $\mathcal{D}$ as $\eta(\mathcal{Z}_*) \sim \mathcal{N}(\mu_{\mathcal{Z}_*|\mathcal{D}}, \Sigma_{\mathcal{Z}_*|\mathcal{D}})$, where

$$\mu_{\mathcal{Z}_*|\mathcal{D}} \triangleq \mu(\mathcal{Z}_*) + k(\mathcal{Z}_*, \mathcal{Z})\mathring{k}(\mathcal{Z}, \mathcal{Z})^{-1}(y - \mu(\mathcal{Z})),$$
$$\Sigma_{\mathcal{Z}_*|\mathcal{D}} \triangleq k(\mathcal{Z}_*, \mathcal{Z}_*) - k(\mathcal{Z}_*, \mathcal{Z})\mathring{k}(\mathcal{Z}, \mathcal{Z})^{-1}k(\mathcal{Z}, \mathcal{Z}_*), \tag{3}$$

with $\mathring{k}(\mathcal{Z}, \mathcal{Z}) \triangleq k(\mathcal{Z}, \mathcal{Z}) + \sigma_e^2 I_{n_s}$. We refer (3) as full GPR. If $y$ is multi-dimensional, GPR is performed for each element.

# 2 Problem formulation

## 2.1 Observation model

Consider a network consisting of $n$ agents and a cloud. We denote the agent set by $\mathcal{V} \triangleq \{1, \ldots, n\}$. At each time instant $t$, the observation model of agent $i$ is given as

$$y^{[i]}(t) = \eta(z^{[i]}(t)) + e^{[i]}(t) \tag{4}$$

where $z^{[i]}(t) \in \mathcal{Z} \subseteq \mathbb{R}^{n_z}$ is the input of $\eta$, $y^{[i]}(t) \in \mathcal{Y} \subseteq \mathbb{R}$ is the observation of the agent, and $e^{[i]}(t)$ is independent zero-mean Gaussian noise with $e^{[i]}(t) \sim \mathcal{N}(0, (\sigma_e^{[i]})^2)$.

## 2.2 Attack model

All communications between the cloud and the agents are attack-free. Some agents are Byzantine agents, and they can behave arbitrarily during learning process, and send arbitrary local predictions to the cloud. We assume that an $\alpha$ fraction of the agents are Byzantine, and the remaining $1 - \alpha$ fraction are benign. We denote $\mathcal{B} \subset \mathcal{V}$ as the set of Byzantine agents, and thus $|\mathcal{B}| = \lfloor \alpha n \rfloor$. Both the cloud and the benign agents are unaware of the identities of Byzantine agents, but we assume the cloud knows that $\frac{1}{4}$ is an upper bound of $\alpha$.

**Assumption 1** *It holds that $0 < \alpha < \frac{1}{4}$.*

Most existing works on Byzantine-tolerant algorithms, e.g., [5], [7], [9], can tolerate Byzantine agents with $\alpha < \frac{1}{2}$. These algorithms only transmit one variable, e.g., the gradients of the loss function or the model parameter, between the cloud and the agents, whereas our federated GPR algorithm requires to exchange two variables, e.g., local mean and variance, to realize the aggregation of predictions.

## 2.3 Objective

The objective of this paper is to design an algorithm, which enables the benign agents and the cloud to correctly learn the function $\eta$ without requiring the agents to share local streaming data $(z^{[i]}(t), y^{[i]}(t))$ despite some agents are subject to Byzantine attacks.

# 3 Byzantine-tolerant federated GPR

In this section, we develop a Byzantine-tolerant federated GPR algorithm to solve the problem formulated in Section 2. This algorithm consists of three modules: agent-based local GPR, cloud-based aggregated GPR and agent-based fused GPR.

---

**Algorithm 1** Byzantine-tolerant federated GPR

---

1: **Input**: test inputs $\mathcal{Z}_*$, prior mean function $\mu$, kernel $k$ and noise variance $(\sigma_e^{[i]})^2$ for $i \in \mathcal{V}$
2: **Initialization**: $\mathcal{D}(0) = \emptyset$.
3: **for** $t = 1, 2, \ldots$ **do**
4:     **for** $i \in \mathcal{V}$ **do**                                                         {Agent-based local GPR}
5:         $\mathcal{D}^{[i]}(t) = \mathcal{D}^{[i]}(t-1) \bigcup (\boldsymbol{z}^{[i]}(t), y^{[i]}(t))$
6:         $\check{\boldsymbol{\mu}}'_{\mathcal{Z}_*|\mathcal{D}^{[i]}(t)}, \check{\boldsymbol{\sigma}}'^2_{\mathcal{Z}_*|\mathcal{D}^{[i]}(t)} = \mathrm{lGPR}(\mathcal{D}^{[i]}(t))$
7:     **end for**
8:     $\hat{\boldsymbol{\mu}}_{\mathcal{Z}_*|\mathcal{D}(t)}, \hat{\boldsymbol{\sigma}}^2_{\mathcal{Z}_*|\mathcal{D}(t)} = \mathrm{cGPR}(\check{\boldsymbol{\mu}}'_{\mathcal{Z}_*|\mathcal{D}^{[i]}(t)}, \check{\boldsymbol{\sigma}}'^2_{\mathcal{Z}_*|\mathcal{D}^{[i]}(t)})$       {Cloud-based aggregated GPR}
9:     **for** $i \in \mathcal{V}$ **do**                                                        {Agent-based fused GPR}
10:         $\tilde{\boldsymbol{\mu}}^{[i]}_{\mathcal{Z}_*|\mathcal{D}(t)}, (\tilde{\boldsymbol{\sigma}}^{[i]}_{\mathcal{Z}_*|\mathcal{D}(t)})^2 = \mathrm{fGPR}(\check{\boldsymbol{\mu}}'_{\mathcal{Z}_*|\mathcal{D}^{[i]}(t)}, \check{\boldsymbol{\sigma}}'^2_{\mathcal{Z}_*|\mathcal{D}^{[i]}(t)}, \hat{\boldsymbol{\mu}}_{\mathcal{Z}_*|\mathcal{D}(t)}, \hat{\boldsymbol{\sigma}}^2_{\mathcal{Z}_*|\mathcal{D}(t)})$
11:     **end for**
12: **end for**

---

Specifically, first at each time instant $t$, each agent $i \in \mathcal{V}$ makes its own prediction of $\eta$ through agent-based local GPR over a given set of points $\mathcal{Z}_* \subseteq \boldsymbol{\mathcal{Z}}$ using local streaming dataset $\mathcal{D}^{[i]}(t) \triangleq (\mathcal{Z}^{[i]}(t), \boldsymbol{y}^{[i]}(t))$ with local input data $\mathcal{Z}^{[i]}(t) \triangleq \{\boldsymbol{z}^{[i]}(1), \ldots, \boldsymbol{z}^{[i]}(t)\}$ and output $\boldsymbol{y}^{[i]}(t) \triangleq [y^{[i]}(1), \ldots, y^{[i]}(t)]^{\mathrm{T}}$. Then the benign agents send correct local predictions to the cloud and the Byzantine agents instead send arbitrary values. Second, the cloud collects all the local predictions and constructs two agent sets by trimming a fraction of outliers with respect to the predictive means and variances. Then the cloud uses a Byzantine-tolerant PoE aggregation rule to compute a global prediction, and broadcasts the global prediction to all the agents. Third, the agents refine the predictions on $\mathcal{Z}_*$ by fusing the predictions from the cloud-based GPR with those from the agent-based local GPR. Note that each agent only communicates local predictions with the cloud, and does not share its local streaming data with the cloud and other agents. The formal description of Byzantine-tolerant federated GPR is given in Algorithm 1.

---

**Algorithm 2** Agent-based local GPR: $\mathrm{lGPR}(\mathcal{D}^{[i]}(t))$

---

1: $\mathcal{Z}^{[i]}(t) = \{\boldsymbol{z}^{[i]}(1), \ldots, \boldsymbol{z}^{[i]}(t)\}$
2: $\boldsymbol{y}^{[i]}(t) = [y^{[i]}(1), \ldots, y^{[i]}(t)]^{\mathrm{T}}$
3: **for** $\boldsymbol{z}_* \in \mathcal{Z}_*$ **do**
4:     Choose $z_*^{[i]}(t) \in \mathrm{proj}(\boldsymbol{z}_*, \mathcal{Z}^{[i]}(t))$
5:     Compute $\check{\mu}'_{\boldsymbol{z}_*|\mathcal{D}^{[i]}(t)} \leftarrow \begin{cases} \check{\mu}_{\boldsymbol{z}_*|\mathcal{D}^{[i]}(t)} & \text{Benign agent,} \\ \star & \text{Byzantine agent} \end{cases}$
6:     Compute $\check{\sigma}'^2_{\boldsymbol{z}_*|\mathcal{D}^{[i]}(t)} \leftarrow \begin{cases} \check{\sigma}^2_{\boldsymbol{z}_*|\mathcal{D}^{[i]}(t)} & \text{Benign agent,} \\ \star & \text{Byzantine agent} \end{cases}$
7:     Send $\check{\mu}'_{\boldsymbol{z}_*|\mathcal{D}^{[i]}(t)}$ and $\check{\sigma}'^2_{\boldsymbol{z}_*|\mathcal{D}^{[i]}(t)}$ to the cloud
8: **end for**
9: **return** $\check{\boldsymbol{\mu}}'_{\mathcal{Z}_*|\mathcal{D}^{[i]}(t)}, \check{\boldsymbol{\sigma}}'^2_{\mathcal{Z}_*|\mathcal{D}^{[i]}(t)}$

---

### 3.1 Agent-based local GPR

To reduce computational complexity, we perform Nearest-neighbor (NN) GPR as the agent-based local GPR to predict for each test point $\boldsymbol{z}_* \in \mathcal{Z}_*$. Instead of feeding the whole training dataset to GPR, the agent-based local GPR only uses the nearest input denoted by $z_*^{[i]}(t) \in \mathrm{proj}(\boldsymbol{z}_*, \mathcal{Z}^{[i]}(t))$ and its corresponding output $y^{[i]}_{z_*^{[i]}(t)}$ to compute the local predictions. Notice that, computation complexity of the hyperparameter tuning for NN GPR is $\mathcal{O}(t^3)$. In fact, the computation complexity of the hyperparameter tuning can be reduced to $\mathcal{O}(1)$ by using recursive hyperparameter tuning [21]. NN GPR in prediction has computation complexity equivalent to nearest-neighbor search, whose worst case complexity is $\mathcal{O}(t)$ [22], while full GPR in (3) has computation complexity $\mathcal{O}(t^3)$ [23]. Denote $\check{\mu}'_{\boldsymbol{z}_*|\mathcal{D}^{[i]}(t)}$ and $\check{\sigma}'^2_{\boldsymbol{z}_*|\mathcal{D}^{[i]}(t)}$ the predictive mean and variance output from the agent-based local GPR. The predictive mean and variance of the benign agents are written in the following forms

$\breve{\mu}'_{\boldsymbol{z}_*|\mathcal{D}^{[i]}(t)} = \breve{\mu}_{\boldsymbol{z}_*|\mathcal{D}^{[i]}(t)}, \breve{\sigma}'^2_{\boldsymbol{z}_*|\mathcal{D}^{[i]}(t)} = \breve{\sigma}^2_{\boldsymbol{z}_*|\mathcal{D}^{[i]}(t)}$ where

$$\breve{\mu}_{\boldsymbol{z}_*|\mathcal{D}^{[i]}(t)} \triangleq \mu(\boldsymbol{z}_*) + k(\boldsymbol{z}_*, \boldsymbol{z}_*^{[i]}(t))\mathring{k}(\boldsymbol{z}_*^{[i]}(t), \boldsymbol{z}_*^{[i]}(t))^{-1}(y^{[i]}_{\boldsymbol{z}_*^{[i]}(t)} - \mu(\boldsymbol{z}_*^{[i]}(t))),$$

$$\breve{\sigma}^2_{\boldsymbol{z}_*|\mathcal{D}^{[i]}(t)} \triangleq k(\boldsymbol{z}_*, \boldsymbol{z}_*) - k(\boldsymbol{z}_*, \boldsymbol{z}_*^{[i]}(t))\mathring{k}(\boldsymbol{z}_*^{[i]}(t), \boldsymbol{z}_*^{[i]}(t))^{-1}k(\boldsymbol{z}_*^{[i]}(t), \boldsymbol{z}_*) \tag{5}$$

with $\mathring{k}(\boldsymbol{z}_*^{[i]}(t), \boldsymbol{z}_*^{[i]}(t)) \triangleq k(\boldsymbol{z}_*^{[i]}(t), \boldsymbol{z}_*^{[i]}(t)) + (\sigma_e^{[i]})^2$. When the agent is a Byzantine agent, its outputs can be arbitrary, and in this case we write $\breve{\mu}'_{\boldsymbol{z}_*|\mathcal{D}^{[i]}(t)} = \star$ and $\breve{\sigma}'^2_{\boldsymbol{z}_*|\mathcal{D}^{[i]}(t)} = \star$. The implementation of the agent-based local GPR is provided in Algorithm 2.

### 3.2 Cloud-based aggregated GPR

In the sequel, we consider a Byzantine-tolerant PoE aggregation rule for the cloud. The algorithm is described as follows:

*Step 1*: For each $\boldsymbol{z}_* \in \mathcal{Z}_*$, the cloud collects potentially corrupted mean $\breve{\mu}'_{\boldsymbol{z}_*|\mathcal{D}^{[i]}(t)}$ and variance $\breve{\sigma}'^2_{\boldsymbol{z}_*|\mathcal{D}^{[i]}(t)}$ from each agent $i$.

*Step 2*: The cloud constructs two agent sets to tolerate Byzantine attacks by removing the largest and smallest $\beta$ fraction of the local predictive means $\breve{\mu}'_{\boldsymbol{z}_*|\mathcal{D}^{[i]}(t)}$ and variances $\breve{\sigma}'^2_{\boldsymbol{z}_*|\mathcal{D}^{[i]}(t)}$, respectively. For $i \in \mathcal{V}$, we sort $\breve{\mu}'_{\boldsymbol{z}_*|\mathcal{D}^{[i]}(t)}$ and $\breve{\sigma}'^2_{\boldsymbol{z}_*|\mathcal{D}^{[i]}(t)}$ in non-descending order. We denote by $\mathcal{T}^\mu_{\max}(t)$ a set of $\lfloor \beta n \rfloor$ agent sets with the largest local predictive means, and by $\mathcal{T}^\mu_{\min}(t)$ a set of $\lfloor \beta n \rfloor$ agents with the smallest local predictive means. Similarly, we denote trimmed sets of variance by $\mathcal{T}^\sigma_{\max}(t)$ and $\mathcal{T}^\sigma_{\min}(t)$. Then, we define

$$\mathcal{M}(t) \triangleq \mathcal{V} \backslash (\mathcal{T}^\mu_{\max}(t) \bigcup \mathcal{T}^\mu_{\min}(t)), \quad \mathcal{C}(t) \triangleq \mathcal{V} \backslash (\mathcal{T}^\sigma_{\max}(t) \bigcup \mathcal{T}^\sigma_{\min}(t)). \tag{6}$$

Notice that $\mathcal{M}(t)$ and $\mathcal{C}(t)$ include $n - 2\lfloor \beta n \rfloor$ agents with extreme values removed. These are the agents whose predictions do not deviate much from the attack-free case.

*Step 3:* Define a global training dataset as $\mathcal{D}(t) \triangleq \bigcup_{i \in \mathcal{V}} \mathcal{D}^{[i]}(t)$, and a common set as $\mathcal{I}(t) \triangleq \mathcal{M}(t) \bigcap \mathcal{C}(t)$. The following lemma shows that $\mathcal{I}(t)$ is never empty.

**Lemma 1** *Suppose that $\beta < \frac{1}{4}$, it holds that $(1 - 4\beta)n \leq |\mathcal{I}(t)| \leq (1 - 2\beta)n$ at each time instant $t$.*

The proof of Lemma 1 can be found in Appendix A.1. The following lemma shows that $\breve{\mu}'_{\boldsymbol{z}_*|\mathcal{D}^{[i]}(t)}$ and $\breve{\sigma}'^2_{\boldsymbol{z}_*|\mathcal{D}^{[i]}(t)}$ for $i \in \mathcal{I}(t)$ are bounded by the values of the benign agents.

---

**Algorithm 3** Cloud-based aggregated GPR: $\text{cGPR}(\breve{\boldsymbol{\mu}}'_{\mathcal{Z}_*|\mathcal{D}^{[i]}(t)}, \breve{\boldsymbol{\sigma}}'^2_{\mathcal{Z}_*|\mathcal{D}^{[i]}(t)})$

---
1: **for** $t = 1, 2 \ldots$ **do**
2:     $\mathcal{D}(t) = \bigcup_{i \in \mathcal{V}} \mathcal{D}^{[i]}(t)$
3:     **for** $\boldsymbol{z}_* \in \mathcal{Z}_*$ **do**
4:         Construct sets $\mathcal{M}(t)$ and $\mathcal{C}(t)$ according to (6)
5:         Compute $\mathcal{I}(t) = \mathcal{M}(t) \bigcap \mathcal{C}(t)$
6:         Implement aggregation rule (7)
7:         Broadcast $\hat{\boldsymbol{\mu}}_{\mathcal{Z}_*|\mathcal{D}(t)}$ and $\hat{\boldsymbol{\sigma}}^2_{\mathcal{Z}_*|\mathcal{D}(t)}$ to all the agents
8:     **end for**
9: **end for**
10: **return** $\hat{\boldsymbol{\mu}}_{\mathcal{Z}_*|\mathcal{D}(t)}, \hat{\boldsymbol{\sigma}}^2_{\mathcal{Z}_*|\mathcal{D}(t)}$

---

**Lemma 2** *Let $0 < \alpha \leq \beta < \frac{1}{4}$ hold. For $i \in \mathcal{I}(t)$, we have that $\min_{j \in \mathcal{V} \backslash \mathcal{B}} \left\{ \breve{\mu}_{\boldsymbol{z}_*|\mathcal{D}^{[j]}(t)} \right\} \leq \breve{\mu}'_{\boldsymbol{z}_*|\mathcal{D}^{[i]}(t)} \leq \max_{j \in \mathcal{V} \backslash \mathcal{B}} \left\{ \breve{\mu}_{\boldsymbol{z}_*|\mathcal{D}^{[j]}(t)} \right\}$, and $\min_{j \in \mathcal{V} \backslash \mathcal{B}} \left\{ \breve{\sigma}^2_{\boldsymbol{z}_*|\mathcal{D}^{[j]}(t)} \right\} \leq \breve{\sigma}'^2_{\boldsymbol{z}_*|\mathcal{D}^{[i]}(t)} \leq \max_{j \in \mathcal{V} \backslash \mathcal{B}} \left\{ \breve{\sigma}^2_{\boldsymbol{z}_*|\mathcal{D}^{[j]}(t)} \right\}$.*

The proof of Lemma 2 is presented in Appendix A.2. Given $\mathcal{I}(t)$, we propose a Byzantine-tolerant PoE aggregation rule below

$$\hat{\mu}_{\boldsymbol{z}_*|\mathcal{D}(t)} \triangleq \frac{\hat{\sigma}^2_{\boldsymbol{z}_*|\mathcal{D}(t)}}{|\mathcal{I}(t)|} \sum_{i\in\mathcal{I}(t)} \breve{\mu}'_{\boldsymbol{z}_*|\mathcal{D}^{[i]}(t)} \breve{\sigma}'^{-2}_{\boldsymbol{z}_*|\mathcal{D}^{[i]}(t)}, \quad \hat{\sigma}^2_{\boldsymbol{z}_*|\mathcal{D}(t)} \triangleq \frac{|\mathcal{I}(t)|}{\sum_{i\in\mathcal{I}(t)} \breve{\sigma}'^{-2}_{\boldsymbol{z}_*|\mathcal{D}^{[i]}(t)}}. \tag{7}$$

Note that if all the agents are known to be benign, we can simply select $\beta = \alpha = 0$, and hence $\mathcal{M}(t) = \mathcal{C}(t) = \mathcal{V}$. In this case, (7) reduces to the standard PoE in [24]. Then the cloud-based aggregated GPR returns $\hat{\boldsymbol{\mu}}_{\mathcal{Z}_*|\mathcal{D}(t)} \triangleq [\hat{\mu}_{\boldsymbol{z}_*|\mathcal{D}(t)}]_{\boldsymbol{z}_*\in\mathcal{Z}_*}$ and $\hat{\boldsymbol{\sigma}}^2_{\mathcal{Z}_*|\mathcal{D}(t)} \triangleq [\hat{\sigma}^2_{\boldsymbol{z}_*|\mathcal{D}(t)}]_{\boldsymbol{z}_*\in\mathcal{Z}_*}$. The implementation of the cloud-based aggregated GPR is provided in Algorithm 3.

### 3.3 Agent-based fused GPR

Notice that the cloud makes predictions using more data than each agent, and it could achieve better learning performances. Then the agent-based fused GPR aims to refine predictions of $\eta(\mathcal{Z}_*)$ by integrating the agent-based local GPR with the cloud-based aggregated GPR. When the cloud variance $\hat{\sigma}'^2_{\boldsymbol{z}_*|\mathcal{D}(t)}$ is lower than the local predictive variances $\breve{\sigma}'^2_{\boldsymbol{z}_*|\mathcal{D}^{[i]}(t)}$, the predictions from the cloud-based aggregated GPR are used. Otherwise, those from the agent-based local GPR are used. Our algorithm is one-round. The refined predictions in the agent-based fused GPR will not be transmitted back to the cloud, hence it cannot change the final aggregation. The implementation of the agent-based fused GPR is provided in Algorithm 4.

---

**Algorithm 4** Agent-based fused GPR: $\text{fGPR}(\breve{\boldsymbol{\mu}}'_{\mathcal{Z}_*|\mathcal{D}^{[i]}(t)}, \breve{\boldsymbol{\sigma}}'^2_{\mathcal{Z}_*|\mathcal{D}^{[i]}(t)}, \hat{\boldsymbol{\mu}}_{\mathcal{Z}_*|\mathcal{D}(t)}, \hat{\boldsymbol{\sigma}}^2_{\mathcal{Z}_*|\mathcal{D}(t)})$

1: **for** $\boldsymbol{z}_* \in \mathcal{Z}_*$ **do**
2:     **if** $\hat{\sigma}^2_{\boldsymbol{z}_*|\mathcal{D}(t)} \leq \breve{\sigma}'^2_{\boldsymbol{z}_*|\mathcal{D}^{[i]}(t)}$ **then**
3:         Compute

$$\tilde{\mu}^{[i]}_{\boldsymbol{z}_*|\mathcal{D}(t)}, (\tilde{\sigma}^{[i]}_{\boldsymbol{z}_*|\mathcal{D}(t)})^2 \leftarrow \begin{cases} \hat{\mu}_{\boldsymbol{z}_*|\mathcal{D}(t)}, \hat{\sigma}^2_{\boldsymbol{z}_*|\mathcal{D}(t)} & \text{Benign agent,} \\ \star & \text{Byzantine agent} \end{cases}$$

4:     **else**
5:         Compute

$$\tilde{\mu}^{[i]}_{\boldsymbol{z}_*|\mathcal{D}(t)}, (\tilde{\sigma}^{[i]}_{\boldsymbol{z}_*|\mathcal{D}(t)})^2 \leftarrow \begin{cases} \breve{\mu}_{\boldsymbol{z}_*|\mathcal{D}^{[i]}(t)}, \breve{\sigma}^2_{\boldsymbol{z}_*|\mathcal{D}^{[i]}(t)} & \text{Benign agent,} \\ \star & \text{Byzantine agent} \end{cases}$$

6:     **end if**
7: **end for**
8: **return** $\tilde{\boldsymbol{\mu}}^{[i]}_{\mathcal{Z}_*|\mathcal{D}(t)}, (\tilde{\boldsymbol{\sigma}}^{[i]}_{\mathcal{Z}_*|\mathcal{D}(t)})^2$

---

### 3.4 Performance guarantee

This section discusses the robustness of the cloud-based aggregated GPR. We start with the assumptions of the kernel function $k$.

**Assumption 2** *1) (Decomposition). The kernel function $k(\cdot, \cdot)$ can be decomposed in such a way that $k(\cdot, \cdot) = \kappa(D(\cdot, \cdot))$, where $\kappa : \mathbb{R}_+ \to \mathbb{R}_{++}$ is continuous.*

*2) (Monotonicity). It holds that $\kappa(s)$ is monotonically decreasing in $s$ and $\kappa(0) = \sigma^2_f$.*

Many existing kernel functions can satisfy Assumption 2, e.g., squared exponential kernel [13]. As [13], any non-zero mean Gaussian process can be decomposed into a deterministic process and a zero-mean stochastic process. Without loss of generality and for notational simplicity, we assume $\eta$ follows a zero-mean Gaussian process in the analysis.

**Assumption 3** *It satisfies that $\eta \sim \mathcal{GP}(0, k)$ and $\eta$ is Lipschitz continuous on $\boldsymbol{\mathcal{Z}}$.*

The target function $\eta$ is completely specified by Gaussian process with kernel function $k$. This assumption is common in the analysis of GPR [25]. The Lipschitz continuity of $\eta$ implies that there

exists some positive constant $\ell_\eta$ such that $|\eta(\boldsymbol{z}) - \eta(\boldsymbol{z}')| \leq \ell_\eta D(\boldsymbol{z}, \boldsymbol{z}')$ for all $\boldsymbol{z}, \boldsymbol{z}' \in \mathcal{Z}$. For $i \in \mathcal{V}$, we define the dispersion of local data as $d^{[i]}(t) \triangleq \sup_{\boldsymbol{z} \in \boldsymbol{\mathcal{Z}}} D(\boldsymbol{z}, \mathcal{Z}^{[i]}(t))$. We denote the maximal local dispersion as $d^{\max}(t) \triangleq \max_{i \in \mathcal{V}}\{d^{[i]}(t)\}$ and sub-Gaussian parameter as $\sigma \triangleq \frac{\sigma_f^2 \sigma_e^{\max}}{\sigma_f^2 + (\sigma_e^{\min})^2}$ where $\sigma_e^{\max} \triangleq \max_{i \in \mathcal{V}}\{\sigma_e^{[i]}\}$ and $\sigma_e^{\min} \triangleq \min_{i \in \mathcal{V}}\{\sigma_e^{[i]}\}$. The following theorem shows that the prediction error from the cloud-based aggregated GPR is upper bounded, and characterizes the bounds of the global predictive variance.

**Theorem 1** *Part I: (Cloud-based aggregated GPR: Mean) Suppose $0 < \alpha \leq \beta < \frac{1}{4}$ and Assumptions 2, 3 hold. For any $\boldsymbol{z}_* \in \boldsymbol{\mathcal{Z}}$ and $0 < \delta < 1$, with probability at least $1 - \delta$, it holds that*
$$\left|\hat{\mu}_{\boldsymbol{z}_*|\mathcal{D}(t)} - \eta(\boldsymbol{z}_*)\right| \leq \left(1 - \frac{\kappa(d^{\max}(t))}{\sigma_f^2 + (\sigma_e^{\max})^2}\right)\|\eta\|_\infty + \frac{\sigma_f^2 \ell_\eta d^{\max}(t)}{\sigma_f^2 + (\sigma_e^{\min})^2} + \sqrt{2\sigma^2(\ln 2 - \ln \delta)} + \Delta(d^{\max}(t))$$
*where* $\Delta(s) \triangleq \frac{2\alpha\left(\sqrt{2\sigma^2(\ln(2n) - \ln \delta)} + \frac{\sigma_f^2 \|\eta\|_\infty}{\sigma_f^2 + (\sigma_e^{\min})^2}\right)}{1 - 4\beta} \frac{\sigma_f^4 + \sigma_f^2(\sigma_e^{\max})^2 - \kappa(s)^2}{\sigma_f^2(\sigma_e^{\min})^2}$.

*Part II: (Cloud-based aggregated GPR: Variance) Let $0 < \alpha \leq \beta < \frac{1}{4}$ and Assumption 2 hold. For any $\boldsymbol{z}_* \in \boldsymbol{\mathcal{Z}}$, it holds that* $\frac{\sigma_f^2(\sigma_e^{\min})^2}{\sigma_f^2 + (\sigma_e^{\max})^2} \leq \hat{\sigma}_{\boldsymbol{z}_*|\mathcal{D}(t)}^2 \leq \sigma_f^2 - \frac{\kappa(d^{\max}(t))^2}{\sigma_f^2 + (\sigma_e^{\max})^2}$.

The proof of Theorem 1 is presented in Appendix A.3. For the attack-free case, i.e., $\alpha = \beta = 0$, the upper bound of Part I becomes $\left(1 - \frac{\kappa(d^{\max}(t))}{\sigma_f^2 + (\sigma_e^{\max})^2}\right)\|\eta\|_\infty + \frac{\sigma_f^2 \ell_\eta d^{\max}(t)}{\sigma_f^2 + (\sigma_e^{\min})^2} + \sqrt{2\sigma^2(\ln 2 - \ln \delta)}$. We assume $\lim_{t \to \infty} d^{[i]}(t) = 0$ for $i \in \mathcal{V}$, the limit of the upper bound is $\frac{(\sigma_e^{\max})^2}{\sigma_f^2 + (\sigma_e^{\max})^2}\|\eta\|_\infty + \sqrt{2\sigma^2(\ln 2 - \ln \delta)}$. Recall that $\sigma = \frac{\sigma_f^2 \sigma_e^{\max}}{\sigma_f^2 + (\sigma_e^{\min})^2}$. Then we can see that $\frac{(\sigma_e^{\max})^2}{\sigma_f^2 + (\sigma_e^{\max})^2}\|\eta\|_\infty$ and $\sqrt{2\sigma^2(\ln 2 - \ln \delta)}$ are positively related to the maximal variance $(\sigma_e^{\max})^2$ of the noise.

The term $\Delta(d^{\max}(t))$ characterizes the performance loss caused by Byzatine agents. First, notice that $\Delta(d^{\max}(t))$ is positively related to $\alpha$, which provides a trade-off between the prediction error and the number of Byzantine agents. With a larger $\alpha$, our algorithm can tolerate more Byzantine agents, but the prediction error gets larger. As $\alpha$ approaches zero, $\Delta(d^{\max}(t))$ goes to zero. This implies that the prediction performance improves if the number of Byzantine agents is decreasing. Second, when $\alpha$ is fixed, $\Delta(d^{\max}(t))$ is proportional to $\beta$. This implies that learning performance improves if the set $\mathcal{I}(t)$ expands.

The upper and lower bounds in Part II are independent of $\alpha$ and $\beta$. Then the variance shares the same upper and lower bounds with that in the attack-free case. This is due to the fact that the extreme values are removed by the Byzantine-tolerant PoE in the cloud-based GPR. As $t$ goes to infinity, we have $\limsup_{t \to \infty} \hat{\sigma}_{\boldsymbol{z}_*|\mathcal{D}(t)}^2 \leq \frac{\sigma_f^2(\sigma_e^{\max})^2}{\sigma_f^2 + (\sigma_e^{\max})^2}$. This implies that the prediction uncertainty diminishes as the variance of measurement noise decreases.

## 4 Numerical experiments

This section experimentally demonstrates the performance of the proposed Byzantine-tolerant federated GPR algorithm using one synthetic dataset and two real-world datasets. We conduct the experiments on a computer with Intel $i7$-6600 CPU, 2.60GHz and 12 GB RAM.

**Synthetic dataset:** For the observation model (4), we consider the target function introduced in [23]
$$\eta(\boldsymbol{z}) = 5\boldsymbol{z}^2 \sin(12\boldsymbol{z}) + (\boldsymbol{z}^3 - 0.5)\sin(3\boldsymbol{z} - 0.5) + 4\cos(2\boldsymbol{z}), \tag{8}$$
with $\boldsymbol{z} \in \mathbb{R}$ and $e \sim \mathcal{N}(0, 0.01)$. We generate $n_s = 10^3, 5 \times 10^3, 10^4, 5 \times 10^4$ training points in $[0, 1]$, respectively, and choose $n_t = 120$ test points randomly in $[0, 1]$. There are $n = 40$ agents in a network. We partition the training dataset into 40 disjoint groups, and each agent is assigned with $n_s/n$ training data points. We use the following squared-exponential kernel $k(\boldsymbol{z}, \boldsymbol{z}_*) = \sigma_f^2 \exp(-\frac{1}{2\ell^2}(\boldsymbol{z} - \boldsymbol{z}_*)^2)$, and use Mean Square Error (MSE) to evaluate the proposed algorithm in terms of consistency, the effect of $\alpha$ and the effect of $\beta$ in the following experiments.

In the first experiment, we let $\alpha = 2.5\%, 5\%, 7.5\%, 10\%, 12.5\%, 15\%$. We randomly choose the agents in the network to be compromised by same-value attacks [26], and let $\beta = \alpha$. Specifically,

for each test point $z_*$, the Byzantine agents only change the local predictive means to 100, that is, $\breve{\mu}'_{z_*|\mathcal{D}^{[i]}(t)} = 100$ for all $t$, and send this incorrect prediction to the cloud. The local predictive variances for all the agents remain unchanged. Fig. 1(a) compares the performances of the proposed Byzantine-tolerant PoE (BtPoE) with the standard PoE. As a benchmark, the bottom curve depicts the prediction errors of the standard PoE (AfPoE) when there is no Byzantine attack in the network; it shows that the standard PoE is consistent in the attack-free scenario. However, when the aforementioned Byzantine attacks are launched, the top curve shows that the standard PoE can no longer remain consistent. Note that there are 6 curves by attacked standard PoE (AedPoE), and only the closest one to the bottom curve is shown in Fig. 1(a). We fix $n_s = 4 \times 10^4$ and $\alpha = 15\%$. The prediction comparisons of these three PoEs are given in Fig. 2, which shows that the learning performance of the Byzantine-tolerant PoE is comparable to that of the attack-free standard PoE with consistency preserved. As Byzantine attacks are launched in the network, the consistency cannot be maintained. Therefore, our proposed algorithm is demonstrated to be resilient to Byzantine attacks.

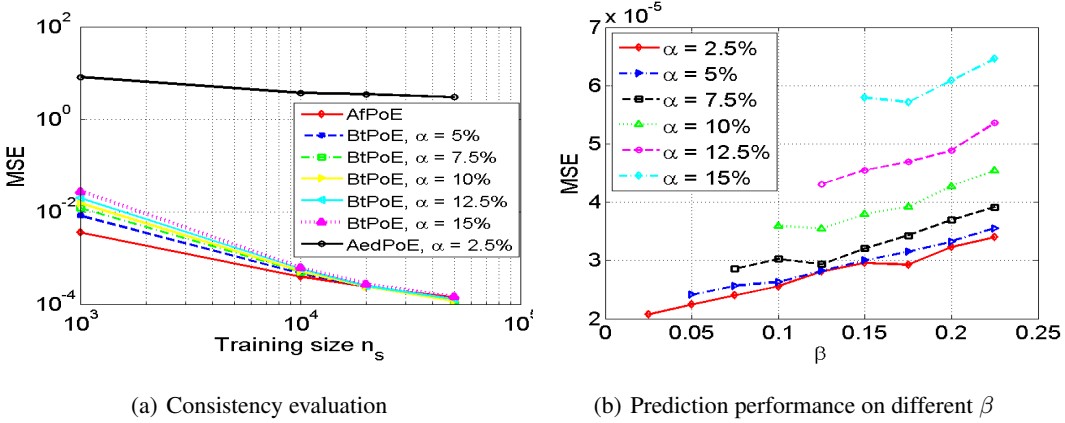

(a) Consistency evaluation         (b) Prediction performance on different $\beta$

Figure 1: Prediction performance of the cloud-based aggregated GPR.

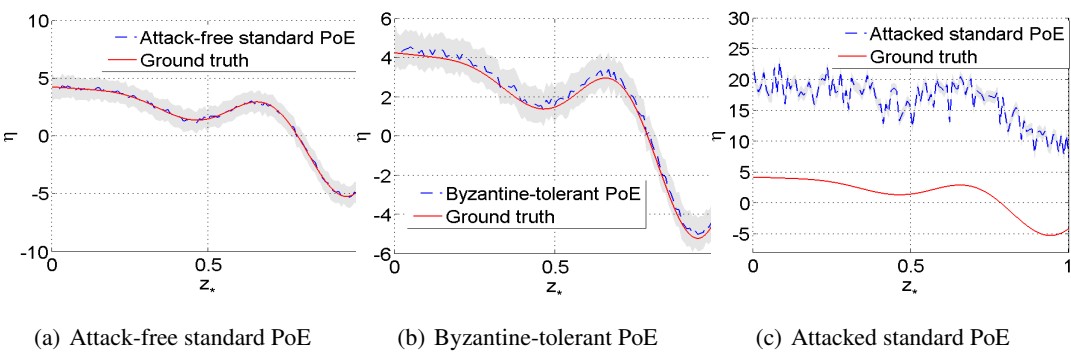

(a) Attack-free standard PoE     (b) Byzantine-tolerant PoE     (c) Attacked standard PoE

Figure 2: Comparisons of the prediction performance in terms of different PoEs. Shaded area represents 99% confidence region.

Notice that the performance of the Byzantine-tolerant PoE increases as $\alpha$, the ratio of Byzantine agents, decreases. Fixing the training data size $n_s = 10^4$, we replicate the experiments for 50 times with different $\alpha$ and observation noise $e$. Figs. 3(a) and 3(b) show that the performances of the Byzantine-tolerant PoE are comparable to that of the attack-free standard PoE with consistency preserved. The prediction errors for both methods are in the order of $10^{-4}$, while Fig. 3(c) shows that the attacked standard PoE has prediction errors ranged from 2.5 to 30. This demonstrates that the proposed PoE aggregation rule has strong ability to tolerate Byzantine attacks.

In the second experiment, we evaluate the prediction performance with regard to $\beta$. All associated parameters are the same as those in the above experiments. Notice that for each $\alpha$, we initially let $\beta = \alpha$ and then increase $\beta$ to 0.225 with increment 0.025, such that the assumption $\alpha \leq \beta < \frac{1}{4}$ is satisfied. The results are shown in Fig. 1(b). This shows that as the number of trimmed agents

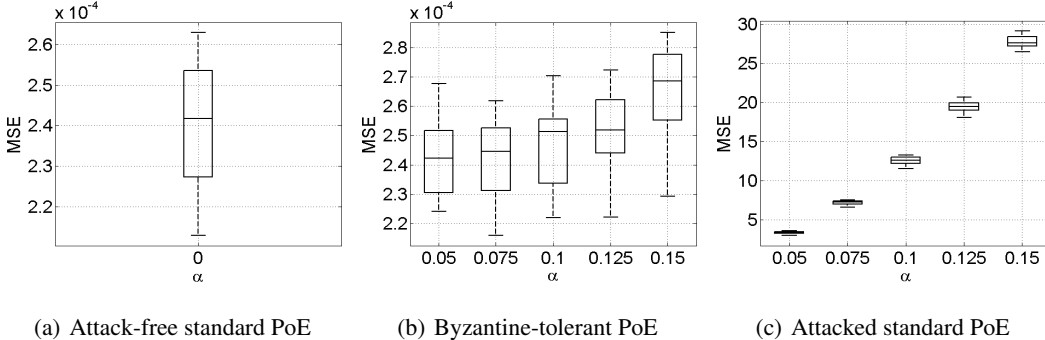

(a) Attack-free standard PoE      (b) Byzantine-tolerant PoE      (c) Attacked standard PoE

Figure 3: Comparison of the prediction performance on different $\alpha$.

increases, the performance of the proposed algorithm grows worse, since more information is excluded in the aggregation. This result validates the conclusion of part I in Theorem 1.

Table 1: Comparison of the prediction performance on different functions.

| Algorithm | Attack-free standard PoE | Byzantine-tolerant PoE | Attacked standard PoE |
|-----------|--------------------------|------------------------|----------------------|
| MSE | $0.0049 \pm 0.007 \times 10^{-3}$ | $0.0236 \pm 0.172 \times 10^{-3}$ | $26.5339 \pm 0.019 \times 10^{-3}$ |

In the third experiment, we conduct Monte Carlo simulations to evaluate the performance of Byzantine-tolerant PoE on different functions. We repeat the experiment for $50$ times by sampling different functions and observation points. Specifically, we replace $\sin(12z)$ in (8) with $\sin(12z) + \epsilon'$ where $\epsilon' \sim \mathcal{N}(0, 0.01i)$, $i = 1, \ldots, 50$. We assign $n = 40$ agents in the network and sample $n_s = 10^4$ training points in $[0, 1]$, and then choose $n_t = 120$ test points randomly in $[0, 1]$. We also use the square-exponential kernel $k(\boldsymbol{z}, \boldsymbol{z}_*)$. There are six Byzantine agents, i.e., $\alpha = 15\%$, and we let $\beta = \alpha$. Experiment results are presented in Table 1, and it shows that the performance of the Byzantine-tolerant PoE is comparable to that of the attack-free standard PoE, noticing that the order of the prediction errors for the Byzantine-tolerant PoE is $10^{-2}$. For comparison, the MSE by the attacked standard PoE is about $26.534$. This demonstrates that the proposed Byzantine-tolerant aggregation rule is effective for learning different functions.

Table 2: Prediction performance comparisons between local GPR and fused GPR on agent 1 and 6.

| Training data size | 6000 | 10000 | 40000 | 150000 |
|--------------------|------|-------|-------|--------|
| MSE/$i = 1$/local GPR | 0.0721 | 0.0681 | 0.0585 | 0.0527 |
| MSE/$i = 1$/fused GPR | 0.0096 | 0.0681 | 0.0094 | 0.0537 |
| MSE/$i = 6$/local GPR | 0.0727 | 0.0566 | 0.0564 | 0.0555 |
| MSE/$i = 6$/fused GPR | 0.0727 | 0.0086 | 0.0564 | 0.0076 |

Moreover, we conduct additional experiments to demonstrate the learning performance improvements by applying our Byzantine-tolerant PoE and the agent-based fused GPR to the synthetic dataset. We assign $n = 40$ agents in the network and sample $n_s = 10^4$ training points in $[0, 1]$, and then choose $n_t = 120$ test points randomly in $[0, 1]$. We also use the square-exponential kernel $k(\boldsymbol{z}, \boldsymbol{z}_*)$. There are six Byzantine agents, i.e., $\alpha = 15\%$, and we let $\beta = \alpha$. The prediction performance comparisons between the agent-based local GPR and fused GPR on agent 1 and 6 are listed in Table 2. For a test point, the local variances $\check{\sigma}'^2_{\boldsymbol{z}_*|\mathcal{D}^{[i]}(t)}$ lie in $[0.3660, 0.6228]$, and the global variance is $\hat{\sigma}^2_{\boldsymbol{z}_*|\mathcal{D}(t)} = 0.4678$. Table 2 shows that the learning performance of the agent-based fused GPR is better than that of the agent-based local GPR. Note that the order of the prediction errors for the agent-based fused GPR can decrease from $10^{-2}$ to $10^{-3}$. This demonstrates that the comparison between variances in step 2 and step 5 in Algorithm 4 is able to improve the learning accuracy, and our algorithm is effective.

**Real-world dataset:** We conduct an experiment to evaluate the prediction performance with different attack magnitudes using two real-world datasets. The first dataset is collected from a seven degrees-of-freedom SARCOS anthropomorphic robot arm [13]. The goal is to learn a function $\eta$. The input of $\eta$

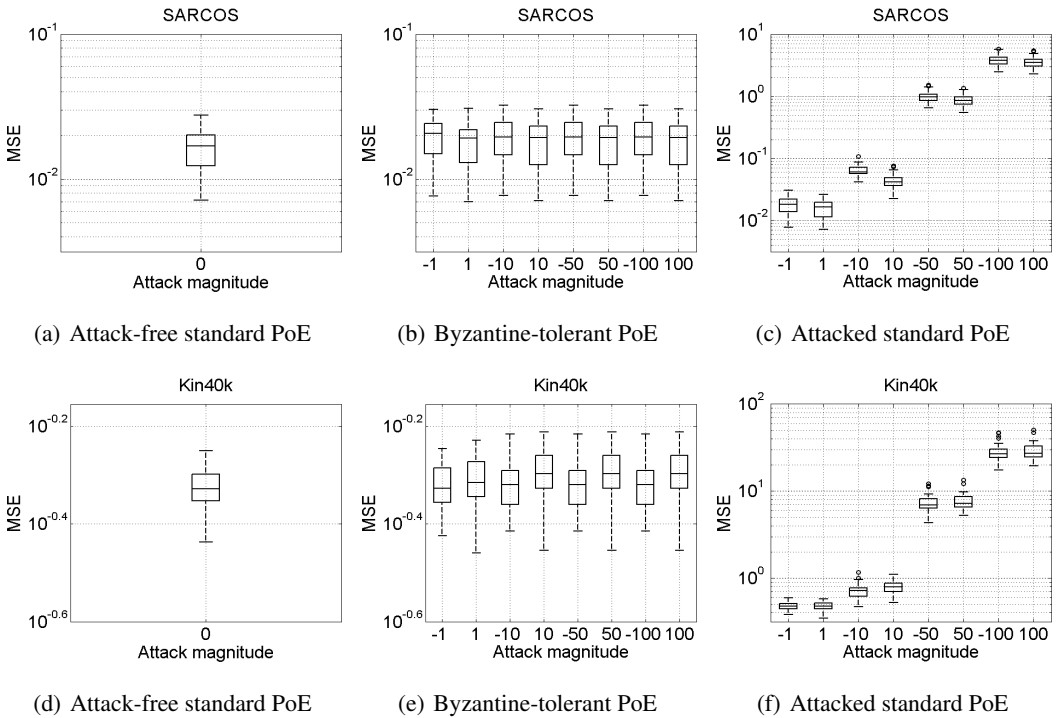

| (a) Attack-free standard PoE | (b) Byzantine-tolerant PoE | (c) Attacked standard PoE |
| (d) Attack-free standard PoE | (e) Byzantine-tolerant PoE | (f) Attacked standard PoE |

Figure 4: Prediction performance with different attack magnitudes on datasets SARCOS and *Kin40k*.

is 21-dimensional including 7 joint positions, 7 joint velocities and 7 joint accelerations; whereas the output is 1-dimensional, representing the torque of a joint. We randomly sample 40,000 data points for training and 40 data points for testing. In the experiment, we partition $40,000$ training data into $n = 40$ disjoint groups, and assign each group to an agent. The second dataset *Kin40k* [27] is created using a robot arm simulator. We randomly choose $9,000$ training points and $40$ test points. We also partition the training data evenly into $40$ disjoint groups. The magnitudes of attack we consider are $\pm 1, \pm 10, \pm 50, \pm 100$. We randomly sample the training data points and repeat the experiments for 50 times. Fig. 4 shows the prediction performances. For both real-world datasets, when the attack magnitude is increasing, the prediction errors of the attacked standard PoE largely increases, whereas those of the attack-free standard PoE and the Byzantine-tolerant PoE are consistently maintained at a low level. It shows that the proposed Byzantine-tolerant PoE is resilient to the attack.

## 5   Conclusion

We propose a Byzantine-tolerant federated GPR algorithm, which is able to tolerate less than one quarter Byzantine agents in the network. We derive the upper bounds on the prediction errors and the lower and upper bounds of the predictive variances. Experiments are conducted to demonstrate the robustness of Byzantine-tolerant GPR algorithm. Future work will focus on the comparison of the learning performance between the agent-based local GPR and the agent-based fused GPR, and the quantification of the benefits the cloud brings to the agents.

## 6   Acknowledgements

This work was partially supported by NSF awards ECCS 1846706 and ECCS 2140175.

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
