# A  Proofs

In this section, we provide complete proofs for each lemma and theorem.

## A.1  Proof of Lemma 1

Notice that $|\mathcal{I}(t)| = |\mathcal{M}(t) \bigcap \mathcal{C}(t)| \leq |\mathcal{M}(t)|$. By definition of $\mathcal{M}(t)$ in (6), we have $\mathcal{M}(t) = \mathcal{V} \setminus (\mathcal{T}_{\max}^{\mu}(t) \bigcup \mathcal{T}_{\min}^{\mu}(t))$. Definitions of $\mathcal{T}_{\max}^{\mu}(t)$ and $\mathcal{T}_{\min}^{\mu}(t)$ give $|\mathcal{T}_{\max}^{\mu}(t)| = |\mathcal{T}_{\min}^{\mu}(t)| = \lfloor \beta n \rfloor$. Then we have $|\mathcal{M}(t)| = |\mathcal{V} \setminus (\mathcal{T}_{\max}^{\mu}(t) \bigcup \mathcal{T}_{\min}^{\mu}(t))| = n - 2\lfloor \beta n \rfloor$, which indicates that $|\mathcal{I}(t)| \leq n - 2\lfloor \beta n \rfloor$. Since $\mathcal{M}(t) \bigcup \mathcal{C}(t) \subseteq \mathcal{V}$, then it holds that $|\mathcal{M}(t) \bigcup \mathcal{C}(t)| \leq n$. By definition of $\mathcal{C}(t)$ in (6), we have $|\mathcal{C}(t)| = n - 2\lfloor \beta n \rfloor$. Therefore, we have $|\mathcal{I}(t)| = |\mathcal{M}(t) \bigcap \mathcal{C}(t)| = |\mathcal{M}(t)| + |\mathcal{C}(t)| - |\mathcal{M}(t) \bigcup \mathcal{C}(t)| \geq n - 4\lfloor \beta n \rfloor$. Since $\beta < \frac{1}{4}$, we have $|\mathcal{I}(t)| > 0$, i.e., $\mathcal{I}(t) \neq \emptyset$.

## A.2  Proof of Lemma 2

Notice that $\mathcal{I}(t) = \mathcal{M}(t) \bigcap \mathcal{C}(t)$. Recall that in Step 2 of Section 3.2, $\check{\mu}'_{\boldsymbol{z}_* | \mathcal{D}^{[i]}(t)}$ is sorted in non-descending order. Without loss of generality, agent 1 has the smallest value and agent $n$ has the largest value, i.e., $\check{\mu}'_{\boldsymbol{z}_* | \mathcal{D}^{[i]}(t)} \leq \check{\mu}'_{\boldsymbol{z}_* | \mathcal{D}^{[i+1]}(t)}$ for $i = 1, 2, \ldots, n-1$. Recall that $\mathcal{T}_{\max}^{\mu}(t)$ contains $\lfloor \beta n \rfloor$ agents with the largest local predictive means. For any $q \in \mathcal{T}_{\max}^{\mu}(t)$ and $q' \in \mathcal{M}(t)$, we have $\check{\mu}'_{\boldsymbol{z}_* | \mathcal{D}^{[q]}(t)} \geq \check{\mu}'_{\boldsymbol{z}_* | \mathcal{D}^{[q']}(t)}$. Suppose that there exists $i \in \mathcal{M}(t)$ such that $\check{\mu}'_{\boldsymbol{z}_* | \mathcal{D}^{[i]}(t)} > \max_{j \in \mathcal{V} \setminus \mathcal{B}} \{ \check{\mu}_{\boldsymbol{z}_* | \mathcal{D}^{[j]}(t)} \}$, then we have $i \in \mathcal{B}$. For all $q \in \mathcal{T}_{\max}^{\mu}(t)$, we have $\check{\mu}'_{\boldsymbol{z}_* | \mathcal{D}^{[q]}(t)} \geq \check{\mu}'_{\boldsymbol{z}_* | \mathcal{D}^{[i]}(t)} > \max_{j \in \mathcal{V} \setminus \mathcal{B}} \{ \check{\mu}_{\boldsymbol{z}_* | \mathcal{D}^{[j]}(t)} \}$. Since $|\mathcal{T}_{\max}^{\mu}(t)| = \lfloor \beta n \rfloor$, then we have $\lfloor \alpha n \rfloor \geq \lfloor \beta n \rfloor + 1$. It contradicts with $\alpha \leq \beta$. Therefore, we have $\check{\mu}'_{\boldsymbol{z}_* | \mathcal{D}^{[i]}(t)} \leq \max_{j \in \mathcal{V} \setminus \mathcal{B}} \{ \check{\mu}_{\boldsymbol{z}_* | \mathcal{D}^{[j]}(t)} \}$ for all $i \in \mathcal{M}(t)$. Likewise, we have $\min_{j \in \mathcal{V} \setminus \mathcal{B}} \{ \check{\mu}_{\boldsymbol{z}_* | \mathcal{D}^{[j]}(t)} \} \leq \check{\mu}'_{\boldsymbol{z}_* | \mathcal{D}^{[i]}(t)}$ for all $i \in \mathcal{M}(t)$.

Analogous to the proof of $\check{\mu}'_{\boldsymbol{z}_* | \mathcal{D}^{[i]}(t)}$, we conclude that $\min_{j \in \mathcal{V} \setminus \mathcal{B}} \{ \check{\sigma}^2_{\boldsymbol{z}_* | \mathcal{D}^{[j]}(t)} \} \leq \check{\sigma}'^2_{\boldsymbol{z}_* | \mathcal{D}^{[i]}(t)} \leq \max_{j \in \mathcal{V} \setminus \mathcal{B}} \{ \check{\sigma}^2_{\boldsymbol{z}_* | \mathcal{D}^{[i]}(t)} \}$ for all $i \in \mathcal{C}(t)$.

## A.3  Proof of Theorem 1

**Part I:** Roadmap of the proof: We first show in Lemma 3 that at time instant $t$, the local predictive mean of agent $i \in \mathcal{V}$ in the attack-free scenario is a sub-Gaussian random variable. Then notice that by triangular inequality, the prediction errors under attacks can be bounded by the magnitude of attacks plus the prediction errors in the attack-free case. Therefore, for $i \in \mathcal{I}(t)$, Lemma 4 uses concentration inequalities of sub-Gaussian random variables to quantify the upper bound of the Byzantine attacks. We derive the upper bound of the prediction error in the attack-free case in Lemma 5.

**Lemma 3** *Let Assumptions 2 and 3 hold. For agent $i \in \mathcal{V}$ and $\boldsymbol{z}_* \in \mathcal{Z}_*$, it holds that $\check{\mu}_{\boldsymbol{z}_* | \mathcal{D}^{[i]}(t)}$ is a sub-Gaussian random variable.*

*Proof:* Pick any $i \in \mathcal{V}$. Monotonicity of Assumption 2 implies that $k(z_*^{[i]}(t), z_*^{[i]}(t)) = \kappa(0) = \sigma_f^2$. By Assumption 3, the prior mean is $\mu(\boldsymbol{z}_*) = \mu(z_*^{[i]}(t)) = 0$. For notational simplicity, we denote the distance by $D_{\boldsymbol{z}_*}^{\mathcal{Z}^{[i]}(t)} \triangleq D(\boldsymbol{z}_*, \mathcal{Z}^{[i]}(t))$. Hence, by (5), the local predictive mean is computed as $\check{\mu}_{\boldsymbol{z}_* | \mathcal{D}^{[i]}(t)} = \frac{\kappa(D_{\boldsymbol{z}_*}^{\mathcal{Z}^{[i]}(t)})}{\sigma_f^2 + (\sigma_e^{[i]})^2} y_{z_*^{[i]}(t)}^{[i]}$. Given the observation model (4), $y_{z_*^{[i]}(t)}^{[i]}$ can be decomposed into a deterministic process $\eta(z_*^{[i]}(t))$ and a zero-mean Gaussian noise $y_{z_*^{[i]}(t)}^{[i]} - \eta(z_*^{[i]}(t))$. For agent $i \in \mathcal{V}$, we denote the expectation and variance of $\check{\mu}_{\boldsymbol{z}_* | \mathcal{D}^{[i]}(t)}$ by $\mathbb{E}\left[ \check{\mu}_{\boldsymbol{z}_* | \mathcal{D}^{[i]}(t)} \right]$ and $Var\left[ \check{\mu}_{\boldsymbol{z}_* | \mathcal{D}^{[i]}(t)} \right]$, respectively. Notice that $y_{z_*^{[i]}(t)}^{[i]} - \eta(z_*^{[i]}(t))$ is the only random variable with variance $(\sigma_e^{[i]})^2$, and this implies $\check{\mu}_{\boldsymbol{z}_* | \mathcal{D}^{[i]}(t)} \sim \mathcal{N}\left( \frac{\kappa(D_{\boldsymbol{z}_*}^{\mathcal{Z}^{[i]}(t)})}{\sigma_f^2 + (\sigma_e^{[i]})^2} \eta(z_*^{[i]}(t)), \left( \frac{\kappa(D_{\boldsymbol{z}_*}^{\mathcal{Z}^{[i]}(t)})}{\sigma_f^2 + (\sigma_e^{[i]})^2} \right)^2 (\sigma_e^{[i]})^2 \right)$. Then for $\lambda_i \in \mathbb{R}$, we conduct the following algebraic calculations

$$\mathbb{E}\left[ \exp(\lambda_i (\check{\mu}_{\boldsymbol{z}_* | \mathcal{D}^{[i]}(t)} - \mathbb{E}\left[ \check{\mu}_{\boldsymbol{z}_* | \mathcal{D}^{[i]}(t)} \right])) \right]$$

$$= \int_{-\infty}^{+\infty} \frac{\exp\left(-\frac{\left(\mu - \mathbb{E}\left[\breve{\mu}_{\boldsymbol{z}_*|\mathcal{D}^{[i]}(t)}\right]\right)^2}{2Var\left[\breve{\mu}_{\boldsymbol{z}_*|\mathcal{D}^{[i]}(t)}\right]}\right)}{\sqrt{2\pi Var\left[\breve{\mu}_{\boldsymbol{z}_*|\mathcal{D}^{[i]}(t)}\right]}} \exp\left(\lambda_i\left(\mu - \mathbb{E}\left[\breve{\mu}_{\boldsymbol{z}_*|\mathcal{D}^{[i]}(t)}\right]\right)\right) d\mu$$

$$= \frac{\exp\left(-\lambda_i \mathbb{E}\left[\breve{\mu}_{\boldsymbol{z}_*|\mathcal{D}^{[i]}(t)}\right]\right)}{\sqrt{2\pi Var\left[\breve{\mu}_{\boldsymbol{z}_*|\mathcal{D}^{[i]}(t)}\right]}} \int_{-\infty}^{+\infty} \exp\left(\lambda_i\mu - \frac{\left(\mu - \mathbb{E}\left[\breve{\mu}_{\boldsymbol{z}_*|\mathcal{D}^{[i]}(t)}\right]\right)^2}{2Var\left[\breve{\mu}_{\boldsymbol{z}_*|\mathcal{D}^{[i]}(t)}\right]}\right) d\mu.$$

Note that

$$\exp\left(-\lambda_i \mathbb{E}\left[\breve{\mu}_{\boldsymbol{z}_*|\mathcal{D}^{[i]}(t)}\right]\right) \exp\left(\lambda_i\mu - \frac{\left(\mu - \mathbb{E}\left[\breve{\mu}_{\boldsymbol{z}_*|\mathcal{D}^{[i]}(t)}\right]\right)^2}{2Var\left[\breve{\mu}_{\boldsymbol{z}_*|\mathcal{D}^{[i]}(t)}\right]}\right)$$

$$= \exp\left(-\lambda_i\left(\mathbb{E}\left[\breve{\mu}_{\boldsymbol{z}_*|\mathcal{D}^{[i]}(t)}\right] - \mu\right) - \frac{\left(\mu - \mathbb{E}\left[\breve{\mu}_{\boldsymbol{z}_*|\mathcal{D}^{[i]}(t)}\right]\right)^2}{2Var\left[\breve{\mu}_{\boldsymbol{z}_*|\mathcal{D}^{[i]}(t)}\right]}\right)$$

$$= \exp\left(\frac{-\left(2\lambda_i\mathbb{E}\left[\breve{\mu}_{\boldsymbol{z}_*|\mathcal{D}^{[i]}(t)}\right] - 2\lambda_i\mu\right)Var\left[\breve{\mu}_{\boldsymbol{z}_*|\mathcal{D}^{[i]}(t)}\right]}{2Var\left[\breve{\mu}_{\boldsymbol{z}_*|\mathcal{D}^{[i]}(t)}\right]}\right)$$

$$\times \exp\left(-\frac{\mu^2 - 2\mu\mathbb{E}\left[\breve{\mu}_{\boldsymbol{z}_*|\mathcal{D}^{[i]}(t)}\right] + \mathbb{E}\left[\breve{\mu}_{\boldsymbol{z}_*|\mathcal{D}^{[i]}(t)}\right]^2}{2Var\left[\breve{\mu}_{\boldsymbol{z}_*|\mathcal{D}^{[i]}(t)}\right]}\right)$$

$$= \exp\left(\frac{\left(\mu - \left(\lambda_iVar\left[\breve{\mu}_{\boldsymbol{z}_*|\mathcal{D}^{[i]}(t)}\right] + \mathbb{E}\left[\breve{\mu}_{\boldsymbol{z}_*|\mathcal{D}^{[i]}(t)}\right]\right)\right)^2}{-2Var\left[\breve{\mu}_{\boldsymbol{z}_*|\mathcal{D}^{[i]}(t)}\right]}\right)$$

$$\times \exp\left(-\lambda_i\mathbb{E}\left[\breve{\mu}_{\boldsymbol{z}_*|\mathcal{D}^{[i]}(t)}\right] - \frac{\mathbb{E}\left[\breve{\mu}_{\boldsymbol{z}_*|\mathcal{D}^{[i]}(t)}\right]^2}{2Var\left[\breve{\mu}_{\boldsymbol{z}_*|\mathcal{D}^{[i]}(t)}\right]}\right) \exp\left(\frac{\left(\lambda_iVar\left[\breve{\mu}_{\boldsymbol{z}_*|\mathcal{D}^{[i]}(t)}\right] + \mathbb{E}\left[\breve{\mu}_{\boldsymbol{z}_*|\mathcal{D}^{[i]}(t)}\right]\right)^2}{2Var\left[\breve{\mu}_{\boldsymbol{z}_*|\mathcal{D}^{[i]}(t)}\right]}\right)$$

$$= \exp\left(\frac{\lambda_i^2 Var\left[\breve{\mu}_{\boldsymbol{z}_*|\mathcal{D}^{[i]}(t)}\right]}{2}\right) \exp\left(\frac{\left(\mu - \left(\lambda_iVar\left[\breve{\mu}_{\boldsymbol{z}_*|\mathcal{D}^{[i]}(t)}\right] + \mathbb{E}\left[\breve{\mu}_{\boldsymbol{z}_*|\mathcal{D}^{[i]}(t)}\right]\right)\right)^2}{-2Var\left[\breve{\mu}_{\boldsymbol{z}_*|\mathcal{D}^{[i]}(t)}\right]}\right).$$

Then we have

$$\mathbb{E}\left[\exp\left(\lambda_i(\breve{\mu}_{\boldsymbol{z}_*|\mathcal{D}^{[i]}(t)} - \mathbb{E}\left[\breve{\mu}_{\boldsymbol{z}_*|\mathcal{D}^{[i]}(t)}\right])\right)\right]$$

$$= \exp\left(\frac{\lambda_i^2 Var\left[\breve{\mu}_{\boldsymbol{z}_*|\mathcal{D}^{[i]}(t)}\right]}{2}\right) \underbrace{\frac{\int_{-\infty}^{+\infty} \exp\left(\frac{\left(\mu - \left(\lambda_iVar\left[\breve{\mu}_{\boldsymbol{z}_*|\mathcal{D}^{[i]}(t)}\right] + \mathbb{E}\left[\breve{\mu}_{\boldsymbol{z}_*|\mathcal{D}^{[i]}(t)}\right]\right)\right)^2}{-2Var\left[\breve{\mu}_{\boldsymbol{z}_*|\mathcal{D}^{[i]}(t)}\right]}\right) d\mu}{\sqrt{2\pi Var\left[\breve{\mu}_{\boldsymbol{z}_*|\mathcal{D}^{[i]}(t)}\right]}}}_{=1}$$

$$= \exp\left(\frac{\lambda_i^2 Var\left[\breve{\mu}_{\boldsymbol{z}_*|\mathcal{D}^{[i]}(t)}\right]}{2}\right). \tag{10}$$

The term $\frac{\exp\left(\frac{\left(\mu - \left(\lambda_iVar\left[\breve{\mu}_{\boldsymbol{z}_*|\mathcal{D}^{[i]}(t)}\right] + \mathbb{E}\left[\breve{\mu}_{\boldsymbol{z}_*|\mathcal{D}^{[i]}(t)}\right]\right)\right)^2}{-2Var\left[\breve{\mu}_{\boldsymbol{z}_*|\mathcal{D}^{[i]}(t)}\right]}\right)}{\sqrt{2\pi Var\left[\breve{\mu}_{\boldsymbol{z}_*|\mathcal{D}^{[i]}(t)}\right]}}$ is a Gaussian probability density function
with mean $\lambda_iVar\left[\breve{\mu}_{\boldsymbol{z}_*|\mathcal{D}^{[i]}(t)}\right] + \mathbb{E}\left[\breve{\mu}_{\boldsymbol{z}_*|\mathcal{D}^{[i]}(t)}\right]$ and variance $Var\left[\breve{\mu}_{\boldsymbol{z}_*|\mathcal{D}^{[i]}(t)}\right]$. Recall that $\sigma^2 = \left(\frac{\sigma_f^2 \sigma_e^{\max}}{\sigma_f^2 + (\sigma_e^{\min})^2}\right)^2$. By Assumption 2, it holds that the kernel function $\kappa(\cdot)$ is monotonically decreasing
and $\kappa(0) = \sigma_f^2$. Therefore, we have $Var\left[\breve{\mu}_{\boldsymbol{z}_*|\mathcal{D}^{[i]}(t)}\right] = \left(\frac{\kappa(D_{\boldsymbol{z}_*}^{\mathcal{Z}^{[i]}(t)})}{\sigma_f^2 + (\sigma_e^{[i]})^2}\sigma_e^{[i]}\right)^2 \le \left(\frac{\sigma_f^2 \sigma_e^{\max}}{\sigma_f^2 + (\sigma_e^{\min})^2}\right)^2 = \sigma^2$.
Substituting $Var\left[\breve{\mu}_{\boldsymbol{z}_*|\mathcal{D}^{[i]}(t)}\right] \le \sigma^2$ into (10) yields $\mathbb{E}\left[\exp\left(\lambda_i(\breve{\mu}_{\boldsymbol{z}_*|\mathcal{D}^{[i]}(t)} - \mathbb{E}\left[\breve{\mu}_{\boldsymbol{z}_*|\mathcal{D}^{[i]}(t)}\right])\right)\right] \le \exp\left(\frac{\lambda_i^2 \sigma^2}{2}\right)$. Thus by Definition 1, we conclude that $\breve{\mu}_{\boldsymbol{z}_*|\mathcal{D}^{[i]}(t)}$ is a sub-Gaussian random variable.
∎

**Lemma 4** *Let $0 < \alpha \leq \beta < \frac{1}{4}$ and Assumption 2 hold. For all $\boldsymbol{z}_* \in \mathcal{Z}_*$ and $0 < \delta < 1$, with probability at least $1 - \delta$, it holds that $\frac{\hat{\sigma}^2_{\boldsymbol{z}_*|\mathcal{D}(t)}}{|\mathcal{I}(t)|} \sum_{i \in \mathcal{I}(t)} \check{\sigma}'^{-2}_{\boldsymbol{z}_*|\mathcal{D}^{[i]}(t)} \left| \check{\mu}'_{\boldsymbol{z}_*|\mathcal{D}^{[i]}(t)} - \check{\mu}_{\boldsymbol{z}_*|\mathcal{D}^{[i]}(t)} \right| \leq$
$\frac{2\alpha\left(\sqrt{2\sigma^2(\ln(2n) - \ln\delta)} + \frac{\sigma_f^2 \|\eta\|_\infty}{\sigma_f^2 + (\sigma_e^{\min})^2}\right)}{1 - 4\beta} \frac{\sigma_f^4 + \sigma_f^2 (\sigma_e^{\max})^2 - \kappa(d^{\max}(t))^2}{\sigma_f^2 (\sigma_e^{\min})^2}.$*

*Proof:* We denote by $\mathcal{F}(t) \triangleq \mathcal{I}(t) \bigcap (\mathcal{V} \backslash \mathcal{B})$ the set of benign agents in the set $\mathcal{I}(t)$. That is, $\check{\mu}'_{\boldsymbol{z}_*|\mathcal{D}^{[i]}(t)} = \check{\mu}_{\boldsymbol{z}_*|\mathcal{D}^{[i]}(t)}$ for all $i \in \mathcal{F}(t)$, therefore it holds that $\frac{\hat{\sigma}^2_{\boldsymbol{z}_*|\mathcal{D}(t)}}{|\mathcal{I}(t)|} \sum_{i \in \mathcal{F}(t)} \check{\sigma}'^{-2}_{\boldsymbol{z}_*|\mathcal{D}^{[i]}(t)} \left| \check{\mu}'_{\boldsymbol{z}_*|\mathcal{D}^{[i]}(t)} - \check{\mu}_{\boldsymbol{z}_*|\mathcal{D}^{[i]}(t)} \right| = 0$. Recall that $\hat{\sigma}^2_{\boldsymbol{z}_*|\mathcal{D}(t)} = \frac{|\mathcal{I}(t)|}{\sum_{j \in \mathcal{I}(t)} \check{\sigma}'^{-2}_{\boldsymbol{z}_*|\mathcal{D}^{[j]}(t)}}$.
We have

$$
\begin{aligned}
&\frac{\hat{\sigma}^2_{\boldsymbol{z}_*|\mathcal{D}(t)}}{|\mathcal{I}(t)|} \sum_{i \in \mathcal{I}(t)} \check{\sigma}'^{-2}_{\boldsymbol{z}_*|\mathcal{D}^{[i]}(t)} \left| \check{\mu}'_{\boldsymbol{z}_*|\mathcal{D}^{[i]}(t)} - \check{\mu}_{\boldsymbol{z}_*|\mathcal{D}^{[i]}(t)} \right| \\
&= \frac{\hat{\sigma}^2_{\boldsymbol{z}_*|\mathcal{D}(t)}}{|\mathcal{I}(t)|} \sum_{i \in \mathcal{I}(t) \bigcap \mathcal{B}} \check{\sigma}'^{-2}_{\boldsymbol{z}_*|\mathcal{D}^{[i]}(t)} \left| \check{\mu}'_{\boldsymbol{z}_*|\mathcal{D}^{[i]}(t)} - \check{\mu}_{\boldsymbol{z}_*|\mathcal{D}^{[i]}(t)} \right| \\
&\quad + \underbrace{\frac{\hat{\sigma}^2_{\boldsymbol{z}_*|\mathcal{D}(t)}}{|\mathcal{I}(t)|} \sum_{i \in \mathcal{F}(t)} \check{\sigma}'^{-2}_{\boldsymbol{z}_*|\mathcal{D}^{[i]}(t)} \left| \check{\mu}'_{\boldsymbol{z}_*|\mathcal{D}^{[i]}(t)} - \check{\mu}_{\boldsymbol{z}_*|\mathcal{D}^{[i]}(t)} \right|}_{=0} \\
&= \frac{\sum_{i \in \mathcal{I}(t) \bigcap \mathcal{B}} \check{\sigma}'^{-2}_{\boldsymbol{z}_*|\mathcal{D}^{[i]}(t)} \left| \check{\mu}'_{\boldsymbol{z}_*|\mathcal{D}^{[i]}(t)} - \check{\mu}_{\boldsymbol{z}_*|\mathcal{D}^{[i]}(t)} \right|}{\sum_{j \in \mathcal{I}(t)} \check{\sigma}'^{-2}_{\boldsymbol{z}_*|\mathcal{D}^{[j]}(t)}} \\
&\leq \frac{\sum_{i \in \mathcal{I}(t) \bigcap \mathcal{B}} \check{\sigma}'^{-2}_{\boldsymbol{z}_*|\mathcal{D}^{[i]}(t)} \left( \left| \check{\mu}'_{\boldsymbol{z}_*|\mathcal{D}^{[i]}(t)} \right| + \left| \check{\mu}_{\boldsymbol{z}_*|\mathcal{D}^{[i]}(t)} \right| \right)}{\sum_{j \in \mathcal{I}(t)} \check{\sigma}'^{-2}_{\boldsymbol{z}_*|\mathcal{D}^{[j]}(t)}}.
\end{aligned} \tag{11}
$$

In the remaining proof, we find the upper bound of $\left| \check{\mu}'_{\boldsymbol{z}_*|\mathcal{D}^{[i]}(t)} \right| + \left| \check{\mu}_{\boldsymbol{z}_*|\mathcal{D}^{[i]}(t)} \right|$, and characterize the lower and upper bounds of $\check{\sigma}'^{-2}_{\boldsymbol{z}_*|\mathcal{D}^{[i]}(t)}$.

1) *The upper bound of* $\left| \check{\mu}'_{\boldsymbol{z}_*|\mathcal{D}^{[i]}(t)} \right| + \left| \check{\mu}_{\boldsymbol{z}_*|\mathcal{D}^{[i]}(t)} \right|$. By Lemma 2, we have $\left| \check{\mu}'_{\boldsymbol{z}_*|\mathcal{D}^{[i]}(t)} \right| \leq \max_{i \in \mathcal{V} \backslash \mathcal{B}} \left\{ \left| \check{\mu}_{\boldsymbol{z}_*|\mathcal{D}^{[i]}(t)} \right| \right\}$ for all $i \in \mathcal{I}(t)$. Then $\left| \check{\mu}'_{\boldsymbol{z}_*|\mathcal{D}^{[i]}(t)} \right| + \left| \check{\mu}_{\boldsymbol{z}_*|\mathcal{D}^{[i]}(t)} \right| \leq 2 \max_{i \in \mathcal{V} \backslash \mathcal{B}} \left\{ \left| \check{\mu}_{\boldsymbol{z}_*|\mathcal{D}^{[i]}(t)} \right| \right\}$ for all $i \in \mathcal{I}(t)$. By Lemma 3, for all $i \in \mathcal{V}$, $\check{\mu}_{\boldsymbol{z}_*|\mathcal{D}^{[i]}(t)}$ is a sub-Gaussian random variable. Since $|\mathcal{V} \backslash \mathcal{B}| = n - \lfloor \alpha n \rfloor$, by maximal inequality (Theorem 1.14 on page 25 in [28]), for any $\epsilon_1 > 0$, we have

$$
P \left\{ \max_{i \in \mathcal{V} \backslash \mathcal{B}} \left\{ \left| \check{\mu}_{\boldsymbol{z}_*|\mathcal{D}^{[i]}(t)} - \frac{\kappa(D_{\boldsymbol{z}_*}^{\mathcal{Z}^{[i]}(t)})}{\sigma_f^2 + (\sigma_e^{[i]})^2} \eta(\boldsymbol{z}_*^{[i]}(t)) \right| \right\} \geq \epsilon_1 \right\} \leq 2(n - \lfloor \alpha n \rfloor) e^{-\frac{\epsilon_1^2}{2\sigma^2}} \leq 2n e^{-\frac{\epsilon_1^2}{2\sigma^2}}.
$$

For $0 < \delta < 1$, choosing $\epsilon_1 \triangleq \sqrt{2\sigma^2(\ln(2n) - \ln\delta)}$, with probability at least $1 - \delta$, we have

$$
\max_{i \in \mathcal{V} \backslash \mathcal{B}} \left\{ \left| \check{\mu}_{\boldsymbol{z}_*|\mathcal{D}^{[i]}(t)} - \frac{\kappa(D_{\boldsymbol{z}_*}^{\mathcal{Z}^{[i]}(t)})}{\sigma_f^2 + (\sigma_e^{[i]})^2} \eta(\boldsymbol{z}_*^{[i]}(t)) \right| \right\} \leq \sqrt{2\sigma^2(\ln(2n) - \ln\delta)}.
$$

Since triangular inequality renders

$$
\left| \check{\mu}_{\boldsymbol{z}_*|\mathcal{D}^{[i]}(t)} \right| \leq \left| \check{\mu}_{\boldsymbol{z}_*|\mathcal{D}^{[i]}(t)} - \frac{\kappa(D_{\boldsymbol{z}_*}^{\mathcal{Z}^{[i]}(t)})}{\sigma_f^2 + (\sigma_e^{[i]})^2} \eta(\boldsymbol{z}_*^{[i]}(t)) \right| + \left| \frac{\kappa(D_{\boldsymbol{z}_*}^{\mathcal{Z}^{[i]}(t)})}{\sigma_f^2 + (\sigma_e^{[i]})^2} \eta(\boldsymbol{z}_*^{[i]}(t)) \right|,
$$

we have

$$
\max_{i \in \mathcal{V} \backslash \mathcal{B}} \left\{ \left| \check{\mu}_{\boldsymbol{z}_*|\mathcal{D}^{[i]}(t)} \right| \right\} \leq \max_{i \in \mathcal{V} \backslash \mathcal{B}} \left\{ \left| \frac{\kappa(D_{\boldsymbol{z}_*}^{\mathcal{Z}^{[i]}(t)})}{\sigma_f^2 + (\sigma_e^{[i]})^2} \eta(\boldsymbol{z}_*^{[i]}(t)) \right| \right\}
$$

$$+ \max_{i \in \mathcal{V} \setminus \mathcal{B}} \left\{ \left| \check{\mu}_{\boldsymbol{z}_* | \mathcal{D}^{[i]}(t)} - \frac{\kappa(D_{\boldsymbol{z}_*}^{\mathcal{Z}^{[i]}(t)})}{\sigma_f^2 + (\sigma_e^{[i]})^2} \eta(\boldsymbol{z}_*^{[i]}(t)) \right| \right\},$$

which implies that with probability at least $1 - \delta$,

$$\max_{i \in \mathcal{V} \setminus \mathcal{B}} \left\{ \left| \check{\mu}_{\boldsymbol{z}_* | \mathcal{D}^{[i]}(t)} \right| \right\} \leq \sqrt{2\sigma^2 (\ln(2n) - \ln \delta)} + \max_{i \in \mathcal{V} \setminus \mathcal{B}} \left\{ \left| \frac{\kappa(D_{\boldsymbol{z}_*}^{\mathcal{Z}^{[i]}(t)})}{\sigma_f^2 + (\sigma_e^{[i]})^2} \eta(\boldsymbol{z}_*^{[i]}(t)) \right| \right\}.$$

Since $|\eta(\boldsymbol{z}_*^{[i]}(t))| \leq \|\eta\|_\infty$, by monotonicity of $\kappa(\cdot)$ in Assumption 2, it holds that $\max_{i \in \mathcal{V} \setminus \mathcal{B}} \left\{ \left| \frac{\kappa(D_{\boldsymbol{z}_*}^{\mathcal{Z}^{[i]}(t)})}{\sigma_f^2 + (\sigma_e^{[i]})^2} \eta(\boldsymbol{z}_*^{[i]}(t)) \right| \right\} \leq \frac{\sigma_f^2 \|\eta\|_\infty}{\sigma_f^2 + (\sigma_e^{\min})^2}$. Then with probability at least $1 - \delta$, we have

$$\max_{i \in \mathcal{V} \setminus \mathcal{B}} \left\{ \left| \check{\mu}_{\boldsymbol{z}_* | \mathcal{D}^{[i]}(t)} \right| \right\} \leq \sqrt{2\sigma^2 (\ln(2n) - \ln \delta)} + \frac{\sigma_f^2 \|\eta\|_\infty}{\sigma_f^2 + (\sigma_e^{\min})^2}.$$

Therefore, with probability at least $1 - \delta$, we have

$$\left| \check{\mu}'_{\boldsymbol{z}_* | \mathcal{D}^{[i]}(t)} \right| + \left| \check{\mu}_{\boldsymbol{z}_* | \mathcal{D}^{[i]}(t)} \right| \leq 2 \max_{i \in \mathcal{V} \setminus \mathcal{B}} \left\{ \left| \check{\mu}_{\boldsymbol{z}_* | \mathcal{D}^{[i]}(t)} \right| \right\}$$

$$\leq 2 \left( \sqrt{2\sigma^2 (\ln(2n) - \ln \delta)} + \frac{\sigma_f^2 \|\eta\|_\infty}{\sigma_f^2 + (\sigma_e^{\min})^2} \right). \tag{12}$$

2) *The lower and upper bounds of* $\check{\sigma}'^{-2}_{\boldsymbol{z}_* | \mathcal{D}^{[i]}(t)}$. Lemma 2 renders that $\min_{j \in \mathcal{V} \setminus \mathcal{B}} \left\{ \check{\sigma}^2_{\boldsymbol{z}_* | \mathcal{D}^{[j]}(t)} \right\} \leq \check{\sigma}'^2_{\boldsymbol{z}_* | \mathcal{D}^{[i]}(t)} \leq \max_{j \in \mathcal{V} \setminus \mathcal{B}} \left\{ \check{\sigma}^2_{\boldsymbol{z}_* | \mathcal{D}^{[j]}(t)} \right\}$ for all $i \in \mathcal{I}(t)$, then we have $\left( \max_{j \in \mathcal{V} \setminus \mathcal{B}} \left\{ \check{\sigma}^2_{\boldsymbol{z}_* | \mathcal{D}^{[j]}(t)} \right\} \right)^{-1} \leq \check{\sigma}'^{-2}_{\boldsymbol{z}_* | \mathcal{D}^{[i]}(t)} \leq \left( \min_{j \in \mathcal{V} \setminus \mathcal{B}} \left\{ \check{\sigma}^2_{\boldsymbol{z}_* | \mathcal{D}^{[j]}(t)} \right\} \right)^{-1}$ for all $i \in \mathcal{I}(t)$. Theorem IV.3 in [29] gives $\frac{\sigma_f^2 (\sigma_e^{[i]})^2}{\sigma_f^2 + (\sigma_e^{[i]})^2} \leq \check{\sigma}^2_{\boldsymbol{z}_* | \mathcal{D}^{[i]}(t)} \leq \sigma_f^2 - \frac{\kappa(d^{[i]}(t))^2}{\sigma_f^2 + (\sigma_e^{[i]})^2}$ for all $\boldsymbol{z}_* \in \mathcal{Z}_*$. By monotonicity of $\kappa(\cdot)$ in Assumption 2, it holds that $\frac{\sigma_f^2 (\sigma_e^{\min})^2}{\sigma_f^2 + (\sigma_e^{\max})^2} \leq \check{\sigma}^2_{\boldsymbol{z}_* | \mathcal{D}^{[i]}(t)} \leq \sigma_f^2 - \frac{\kappa(d^{\max}(t))^2}{\sigma_f^2 + (\sigma_e^{\max})^2}$, which implies that for any $i \in \mathcal{I}(t), \boldsymbol{z}_* \in \mathcal{Z}_*$,

$$\frac{\sigma_f^2 + (\sigma_e^{\max})^2}{\sigma_f^4 + \sigma_f^2 (\sigma_e^{\max})^2 - \kappa(d^{\max}(t))^2} \leq \check{\sigma}'^{-2}_{\boldsymbol{z}_* | \mathcal{D}^{[i]}(t)} \leq \frac{\sigma_f^2 + (\sigma_e^{\max})^2}{\sigma_f^2 (\sigma_e^{\min})^2}.$$

Lemma 1 shows that $|\mathcal{I}(t)| \geq n - 4\lfloor \beta n \rfloor$. Since $\lfloor \beta n \rfloor \leq \beta n$, it indicates that $|\mathcal{I}(t)| \geq (1 - 4\beta)n$. Then we have

$$\sum_{j \in \mathcal{I}(t)} \check{\sigma}'^{-2}_{\boldsymbol{z}_* | \mathcal{D}^{[j]}(t)} \geq \frac{(1 - 4\beta)n \left( \sigma_f^2 + (\sigma_e^{\max})^2 \right)}{\sigma_f^4 + \sigma_f^2 (\sigma_e^{\max})^2 - \kappa(d^{\max}(t))^2}. \tag{13}$$

Since $|\mathcal{I}(t) \bigcap \mathcal{B}| \leq |\mathcal{B}| = \lfloor \alpha n \rfloor \leq \alpha n$, by (12), with probability at least $1 - \delta$, we have

$$\sum_{i \in \mathcal{I}(t) \bigcap \mathcal{B}} \check{\sigma}'^{-2}_{\boldsymbol{z}_* | \mathcal{D}^{[i]}(t)} \left( \left| \check{\mu}'_{\boldsymbol{z}_* | \mathcal{D}^{[i]}(t)} \right| + \left| \check{\mu}_{\boldsymbol{z}_* | \mathcal{D}^{[i]}(t)} \right| \right)$$

$$\leq 2\alpha n \left( \sqrt{2\sigma^2 (\ln(2n) - \ln \delta)} + \frac{\sigma_f^2 \|\eta\|_\infty}{\sigma_f^2 + (\sigma_e^{\min})^2} \right) \frac{\sigma_f^2 + (\sigma_e^{\max})^2}{\sigma_f^2 (\sigma_e^{\min})^2}. \tag{14}$$

Combining (13) and (14) with (11) renders that with probability at least $1 - \delta$, it holds that

$$\frac{\hat{\sigma}^2_{\boldsymbol{z}_* | \mathcal{D}(t)}}{|\mathcal{I}(t)|} \sum_{i \in \mathcal{I}(t)} \check{\sigma}'^{-2}_{\boldsymbol{z}_* | \mathcal{D}^{[i]}(t)} \left| \check{\mu}'_{\boldsymbol{z}_* | \mathcal{D}^{[i]}(t)} - \check{\mu}_{\boldsymbol{z}_* | \mathcal{D}^{[i]}(t)} \right|$$

$$\leq \frac{2\alpha \left( \sqrt{2\sigma^2 (\ln(2n) - \ln \delta)} + \frac{\sigma_f^2 \|\eta\|_\infty}{\sigma_f^2 + (\sigma_e^{\min})^2} \right)}{1 - 4\beta} \frac{\sigma_f^4 + \sigma_f^2 (\sigma_e^{\max})^2 - \kappa(d^{\max}(t))^2}{\sigma_f^2 (\sigma_e^{\min})^2}.$$

■

The following Lemma characterizes the upper bound of the prediction error in the attack-free case.

**Lemma 5** *Suppose Assumptions 2 and 3 hold. For $\boldsymbol{z}_* \in \mathcal{Z}_*$, with probability at least $1-\delta$, it holds that*
$$\frac{\hat{\sigma}^2_{\boldsymbol{z}_*|\mathcal{D}(t)}}{|\mathcal{I}(t)|} \sum_{i \in \mathcal{I}(t)} \breve{\sigma}'^{-2}_{\boldsymbol{z}_*|\mathcal{D}^{[i]}(t)} \left| \breve{\mu}_{\boldsymbol{z}_*|\mathcal{D}^{[i]}(t)} - \eta(\boldsymbol{z}_*) \right| \leq (1 - \frac{\kappa(d^{\max}(t))}{\sigma_f^2 + (\sigma_e^{\max})^2}) \|\eta\|_\infty + \frac{\sigma_f^2}{\sigma_f^2 + (\sigma_e^{\min})^2} \ell_\eta d^{\max}(t) + \sqrt{2\sigma^2(\ln 2 - \ln \delta)}.$$

*Proof:* Recall that $\breve{\mu}_{\boldsymbol{z}_*|\mathcal{D}^{[i]}(t)} = \frac{\kappa(D_{\boldsymbol{z}_*}^{\mathcal{Z}^{[i]}(t)})}{\sigma_f^2 + (\sigma_e^{[i]})^2} y^{[i]}_{\boldsymbol{z}_*^{[i]}(t)}$. Then for $i \in \mathcal{V}$, we have

$$\breve{\mu}_{\boldsymbol{z}_*|\mathcal{D}^{[i]}(t)} - \eta(\boldsymbol{z}_*) = (1 - \frac{\kappa(D_{\boldsymbol{z}_*}^{\mathcal{Z}^{[i]}(t)})}{\sigma_f^2 + (\sigma_e^{[i]})^2})(-\eta(\boldsymbol{z}_*)) + \frac{\kappa(D_{\boldsymbol{z}_*}^{\mathcal{Z}^{[i]}(t)})}{\sigma_f^2 + (\sigma_e^{[i]})^2}(y^{[i]}_{\boldsymbol{z}_*^{[i]}(t)} - \eta(\boldsymbol{z}_*^{[i]}(t)))$$
$$+ \frac{\kappa(D_{\boldsymbol{z}_*}^{\mathcal{Z}^{[i]}(t)})}{\sigma_f^2 + (\sigma_e^{[i]})^2}(\eta(\boldsymbol{z}_*^{[i]}(t)) - \eta(\boldsymbol{z}_*)).$$

By triangular inequality, we have

$$|\breve{\mu}_{\boldsymbol{z}_*|\mathcal{D}^{[i]}(t)} - \eta(\boldsymbol{z}_*)| \leq \frac{\kappa(D_{\boldsymbol{z}_*}^{\mathcal{Z}^{[i]}(t)})}{\sigma_f^2 + (\sigma_e^{[i]})^2}|\eta(\boldsymbol{z}_*^{[i]}(t)) - \eta(\boldsymbol{z}_*)| + \frac{\kappa(D_{\boldsymbol{z}_*}^{\mathcal{Z}^{[i]}(t)})}{\sigma_f^2 + (\sigma_e^{[i]})^2}|y^{[i]}_{\boldsymbol{z}_*^{[i]}(t)} - \eta(\boldsymbol{z}_*^{[i]}(t))|$$
$$+ (1 - \frac{\kappa(D_{\boldsymbol{z}_*}^{\mathcal{Z}^{[i]}(t)})}{\sigma_f^2 + (\sigma_e^{[i]})^2})|\eta(\boldsymbol{z}_*)|. \tag{15}$$

We analyze the upper bound of each term on the right-hand side of the inequality (15).

*Term 1.* Recall that $\boldsymbol{z}_*^{[i]}(t) \in \text{proj}(\boldsymbol{z}_*, \mathcal{Z}^{[i]}(t))$. The Lipschitz continuity of $\eta$ in Assumption 3 gives

$$\frac{\kappa(D_{\boldsymbol{z}_*}^{\mathcal{Z}^{[i]}(t)})}{\sigma_f^2 + (\sigma_e^{[i]})^2} \left| \eta(\boldsymbol{z}_*^{[i]}(t)) - \eta(\boldsymbol{z}_*) \right| \leq \frac{\kappa(D_{\boldsymbol{z}_*}^{\mathcal{Z}^{[i]}(t)})}{\sigma_f^2 + (\sigma_e^{[i]})^2} \ell_\eta D_{\boldsymbol{z}_*}^{\mathcal{Z}^{[i]}(t)} \leq \frac{\sigma_f^2}{\sigma_f^2 + (\sigma_e^{\min})^2} \ell_\eta d^{\max}(t). \tag{16}$$

*Term 2.* Recall that $\breve{\mu}_{\boldsymbol{z}_*|\mathcal{D}^{[i]}(t)}$ follows a Gaussian probability distribution and $\breve{\mu}_{\boldsymbol{z}_*|\mathcal{D}^{[i]}(t)} \sim$ $\mathcal{N}\left( \frac{\kappa(D_{\boldsymbol{z}_*}^{\mathcal{Z}^{[i]}(t)})}{\sigma_f^2 + (\sigma_e^{[i]})^2}\eta(\boldsymbol{z}_*^{[i]}(t)), \left(\frac{\kappa(D_{\boldsymbol{z}_*}^{\mathcal{Z}^{[i]}(t)})}{\sigma_f^2 + (\sigma_e^{[i]})^2}\right)^2 (\sigma_e^{[i]})^2 \right)$. By Lemma 3, for all $i \in \mathcal{V}$, we have that $\breve{\mu}_{\boldsymbol{z}_*|\mathcal{D}^{[i]}(t)}$ is a sub-Gaussian random variable. Then by concentration inequality of the sub-Gaussian random variable (see Lemma 1.3 of [28]), for any $\epsilon_2 > 0$, we have

$$P\left\{ \left| \breve{\mu}_{\boldsymbol{z}_*|\mathcal{D}^{[i]}(t)} - \frac{\kappa(D_{\boldsymbol{z}_*}^{\mathcal{Z}^{[i]}(t)})}{\sigma_f^2 + (\sigma_e^{[i]})^2}\eta(\boldsymbol{z}_*^{[i]}(t)) \right| > \epsilon_2 \right\} \leq 2e^{-\frac{\epsilon_2^2}{2\sigma^2}}.$$

Combining the above inequality with $\breve{\mu}_{\boldsymbol{z}_*|\mathcal{D}^{[i]}(t)} = \frac{\kappa(D_{\boldsymbol{z}_*}^{\mathcal{Z}^{[i]}(t)})}{\sigma_f^2 + (\sigma_e^{[i]})^2} y^{[i]}_{\boldsymbol{z}_*^{[i]}(t)}$, for $0 < \delta < 1$, choosing $\epsilon_2 \triangleq \sqrt{2\sigma^2(\ln 2 - \ln \delta)}$, with probability at least $1 - \delta$, it holds

$$\frac{\kappa(D_{\boldsymbol{z}_*}^{\mathcal{Z}^{[i]}(t)})}{\sigma_f^2 + (\sigma_e^{[i]})^2}|y^{[i]}_{\boldsymbol{z}_*^{[i]}(t)} - \eta(\boldsymbol{z}_*^{[i]}(t))| = \left| \breve{\mu}_{\boldsymbol{z}_*|\mathcal{D}^{[i]}(t)} - \frac{\kappa(D_{\boldsymbol{z}_*}^{\mathcal{Z}^{[i]}(t)})}{\sigma_f^2 + (\sigma_e^{[i]})^2}\eta(\boldsymbol{z}_*^{[i]}(t)) \right| \leq \sqrt{2\sigma^2(\ln 2 - \ln \delta)}. \tag{17}$$

*Term 3.* We have $|\eta(\boldsymbol{z}_*)| \leq \|\eta\|_\infty$. By monotonicity of $\kappa(\cdot)$ in Assumption 2, it gives

$$(1 - \frac{\kappa(D_{\boldsymbol{z}_*}^{\mathcal{Z}^{[i]}(t)})}{\sigma_f^2 + (\sigma_e^{[i]})^2})|\eta(\boldsymbol{z}_*)| \leq (1 - \frac{\kappa(d^{\max}(t))}{\sigma_f^2 + (\sigma_e^{\max})^2})\|\eta\|_\infty. \tag{18}$$

Therefore, applying the inequalities (16), (17) and (18) to (15), for $0 < \delta < 1$, with probability at least $1 - \delta$, we have that for all $i \in \mathcal{I}(t)$,

$$\left|\check{\mu}_{\boldsymbol{z}_*|\mathcal{D}^{[i]}(t)} - \eta(\boldsymbol{z}_*)\right| \le (1 - \frac{\kappa(d^{\max}(t))}{\sigma_f^2 + (\sigma_e^{\max})^2})\|\eta\|_\infty + \frac{\sigma_f^2 \ell_\eta d^{\max}(t)}{\sigma_f^2 + (\sigma_e^{\min})^2} + \sqrt{2\sigma^2(\ln 2 - \ln \delta)}. \quad (19)$$

By (7), we have $0 < \frac{\hat{\sigma}_{\boldsymbol{z}_*|\mathcal{D}(t)}^2}{|\mathcal{I}(t)|}\check{\sigma}'^{-2}_{\boldsymbol{z}_*|\mathcal{D}^{[i]}(t)} < 1$ and $\frac{\hat{\sigma}_{\boldsymbol{z}_*|\mathcal{D}(t)}^2}{|\mathcal{I}(t)|}\sum_{i \in \mathcal{I}(t)}\check{\sigma}'^{-2}_{\boldsymbol{z}_*|\mathcal{D}^{[i]}(t)} = 1$, which implies that with probability at least $1 - \delta$, it holds that $\frac{\hat{\sigma}_{\boldsymbol{z}_*|\mathcal{D}(t)}^2}{|\mathcal{I}(t)|}\sum_{i \in \mathcal{I}(t)}\check{\sigma}'^{-2}_{\boldsymbol{z}_*|\mathcal{D}^{[i]}(t)}\left|\check{\mu}_{\boldsymbol{z}_*|\mathcal{D}^{[i]}(t)} - \eta(\boldsymbol{z}_*)\right| \le (1 - \frac{\kappa(d^{\max}(t))}{\sigma_f^2 + (\sigma_e^{\max})^2})\|\eta\|_\infty + \frac{\sigma_f^2 \ell_\eta d^{\max}(t)}{\sigma_f^2 + (\sigma_e^{\min})^2} + \sqrt{2\sigma^2(\ln 2 - \ln \delta)}$. ∎

With Lemmas 3, 4 and 5, we now proceed to complete the proof of part I in Theorem 1.

*Proof of part I in Theorem 1:* Note that, given (7), we have

$$\left|\hat{\mu}_{\boldsymbol{z}_*|\mathcal{D}(t)} - \eta(\boldsymbol{z}_*)\right| = \left|\frac{\hat{\sigma}_{\boldsymbol{z}_*|\mathcal{D}(t)}^2}{|\mathcal{I}(t)|}\sum_{i \in \mathcal{I}(t)}\check{\mu}'_{\boldsymbol{z}_*|\mathcal{D}^{[i]}(t)}\check{\sigma}'^{-2}_{\boldsymbol{z}_*|\mathcal{D}^{[i]}(t)} - \eta(\boldsymbol{z}_*)\right|$$

$$= \left|\frac{\hat{\sigma}_{\boldsymbol{z}_*|\mathcal{D}(t)}^2}{|\mathcal{I}(t)|}\sum_{i \in \mathcal{I}(t)}\check{\mu}'_{\boldsymbol{z}_*|\mathcal{D}^{[i]}(t)}\check{\sigma}'^{-2}_{\boldsymbol{z}_*|\mathcal{D}^{[i]}(t)} - \frac{\hat{\sigma}_{\boldsymbol{z}_*|\mathcal{D}(t)}^2}{|\mathcal{I}(t)|}\sum_{i \in \mathcal{I}(t)}\check{\mu}_{\boldsymbol{z}_*|\mathcal{D}^{[i]}(t)}\check{\sigma}'^{-2}_{\boldsymbol{z}_*|\mathcal{D}^{[i]}(t)}\right.$$

$$\left. + \frac{\hat{\sigma}_{\boldsymbol{z}_*|\mathcal{D}(t)}^2}{|\mathcal{I}(t)|}\sum_{i \in \mathcal{I}(t)}\check{\mu}_{\boldsymbol{z}_*|\mathcal{D}^{[i]}(t)}\check{\sigma}'^{-2}_{\boldsymbol{z}_*|\mathcal{D}^{[i]}(t)} - \eta(\boldsymbol{z}_*)\right|. \quad (20)$$

Since $\frac{\hat{\sigma}_{\boldsymbol{z}_*|\mathcal{D}(t)}^2}{|\mathcal{I}(t)|}\sum_{i \in \mathcal{I}(t)}\check{\sigma}'^{-2}_{\boldsymbol{z}_*|\mathcal{D}^{[i]}(t)} = 1$, then this implies that $\frac{\hat{\sigma}_{\boldsymbol{z}_*|\mathcal{D}(t)}^2}{|\mathcal{I}(t)|}\sum_{i \in \mathcal{I}(t)}\check{\mu}_{\boldsymbol{z}_*|\mathcal{D}^{[i]}(t)}\check{\sigma}'^{-2}_{\boldsymbol{z}_*|\mathcal{D}^{[i]}(t)} - \eta(\boldsymbol{z}_*) = \frac{\hat{\sigma}_{\boldsymbol{z}_*|\mathcal{D}(t)}^2}{|\mathcal{I}(t)|}\sum_{i \in \mathcal{I}(t)}\check{\sigma}'^{-2}_{\boldsymbol{z}_*|\mathcal{D}^{[i]}(t)}\left(\check{\mu}_{\boldsymbol{z}_*|\mathcal{D}^{[i]}(t)} - \eta(\boldsymbol{z}_*)\right)$. Therefore, by triangular inequality, (20) is upper bounded as

$$\left|\frac{\hat{\sigma}_{\boldsymbol{z}_*|\mathcal{D}(t)}^2}{|\mathcal{I}(t)|}\sum_{i \in \mathcal{I}(t)}\check{\mu}'_{\boldsymbol{z}_*|\mathcal{D}^{[i]}(t)}\check{\sigma}'^{-2}_{\boldsymbol{z}_*|\mathcal{D}^{[i]}(t)} - \eta(\boldsymbol{z}_*)\right|$$

$$\le \frac{\hat{\sigma}_{\boldsymbol{z}_*|\mathcal{D}(t)}^2}{|\mathcal{I}(t)|}\sum_{i \in \mathcal{I}(t)}\check{\sigma}'^{-2}_{\boldsymbol{z}_*|\mathcal{D}^{[i]}(t)}\left|\check{\mu}_{\boldsymbol{z}_*|\mathcal{D}^{[i]}(t)} - \eta(\boldsymbol{z}_*)\right|$$

$$+ \frac{\hat{\sigma}_{\boldsymbol{z}_*|\mathcal{D}(t)}^2}{|\mathcal{I}(t)|}\sum_{i \in \mathcal{I}(t)}\check{\sigma}'^{-2}_{\boldsymbol{z}_*|\mathcal{D}^{[i]}(t)}\left|\check{\mu}'_{\boldsymbol{z}_*|\mathcal{D}^{[i]}(t)} - \check{\mu}_{\boldsymbol{z}_*|\mathcal{D}^{[i]}(t)}\right|.$$

Then, combining this with Lemmas 4 and 5, we complete the proof of part I.

**Part II:** We give the upper bound and lower bound of $\hat{\sigma}_{\boldsymbol{z}_*|\mathcal{D}(t)}^2$ as follows:

1) Upper bound. Recall that $\hat{\sigma}_{\boldsymbol{z}_*|\mathcal{D}(t)}^2 = \frac{|\mathcal{I}(t)|}{\sum_{i \in \mathcal{I}(t)}\check{\sigma}'^{-2}_{\boldsymbol{z}_*|\mathcal{D}^{[i]}(t)}}$. Note that $f(x) = \frac{1}{x}$ is a convex function for $x > 0$. By Jensen's inequality (see page 21 in [30]), we have $f(\frac{1}{n}\sum_{i=1}^n x_i) \le \frac{1}{n}\sum_{i=1}^n f(x_i)$. Then plugging in $x_i = \check{\sigma}'^{-2}_{\boldsymbol{z}_*|\mathcal{D}^{[i]}(t)}$, we have

$$\hat{\sigma}_{\boldsymbol{z}_*|\mathcal{D}(t)}^2 = \frac{|\mathcal{I}(t)|}{\sum_{i \in \mathcal{I}(t)}\check{\sigma}'^{-2}_{\boldsymbol{z}_*|\mathcal{D}^{[i]}(t)}} = f(\frac{\sum_{i \in \mathcal{I}(t)}\check{\sigma}'^{-2}_{\boldsymbol{z}_*|\mathcal{D}^{[i]}(t)}}{|\mathcal{I}(t)|}) \le \frac{1}{|\mathcal{I}(t)|}\sum_{i \in \mathcal{I}(t)}\check{\sigma}'^2_{\boldsymbol{z}_*|\mathcal{D}^{[i]}(t)}.$$

It suffices to show the upper bound of $\sum_{i \in \mathcal{I}(t)}\check{\sigma}'^2_{\boldsymbol{z}_*|\mathcal{D}^{[i]}(t)}$. We decompose the agent set $\mathcal{I}(t)$ into two subsets $\mathcal{F}(t)$ and $\mathcal{I}(t)\bigcap\mathcal{B}$ where $\mathcal{F}(t) \triangleq \mathcal{I}(t)\bigcap(\mathcal{V}\backslash\mathcal{B})$ contains the benign agents and $\mathcal{I}(t)\bigcap\mathcal{B}$ contains the Byzantine agents, then we have

$$\sum_{i \in \mathcal{I}(t)}\check{\sigma}'^2_{\boldsymbol{z}_*|\mathcal{D}^{[i]}(t)} = \sum_{i \in \mathcal{F}(t)}\check{\sigma}'^2_{\boldsymbol{z}_*|\mathcal{D}^{[i]}(t)} + \sum_{i \in \mathcal{I}(t)\bigcap\mathcal{B}}\check{\sigma}'^2_{\boldsymbol{z}_*|\mathcal{D}^{[i]}(t)}. \quad (21)$$

We proceed to analyze each term on the right-hand side of (21).

First, notice that $\mathcal{F}(t)$ is the set of benign agents, hence it holds that $\check{\sigma}'^2_{\boldsymbol{z}_*|\mathcal{D}^{[i]}(t)} = \check{\sigma}^2_{\boldsymbol{z}_*|\mathcal{D}^{[i]}(t)}$ for all $i \in \mathcal{F}(t)$. According to Theorem IV.3 in [29], we have $\check{\sigma}^2_{\boldsymbol{z}_*|\mathcal{D}^{[i]}(t)} \leq \sigma^2_f - \frac{\kappa(d^{[i]}(t))^2}{\sigma^2_f + (\sigma^{[i]}_e)^2}$ for all $\boldsymbol{z}_* \in \mathcal{Z}_*$. Monotonicity of $\kappa(\cdot)$ in Assumption 2 shows that $\kappa(d^{\max}(t))^2 \leq \kappa(d^{[i]}(t))^2$, which indicates $\frac{\kappa(d^{\max}(t))^2}{\sigma^2_f + (\sigma^{\max}_e)^2} \leq \frac{\kappa(d^{[i]}(t))^2}{\sigma^2_f + (\sigma^{[i]}_e)^2}$. Therefore, we have

$$\sum_{i \in \mathcal{F}(t)} \check{\sigma}'^2_{\boldsymbol{z}_*|\mathcal{D}^{[i]}(t)} \leq \sum_{i \in \mathcal{F}(t)} \left( \sigma^2_f - \frac{\kappa(d^{[i]}(t))^2}{\sigma^2_f + (\sigma^{[i]}_e)^2} \right) \leq |\mathcal{F}(t)| \left( \sigma^2_f - \frac{\kappa(d^{\max}(t))^2}{\sigma^2_f + (\sigma^{\max}_e)^2} \right). \tag{22}$$

Second, by Lemma 2, we have

$$\sum_{i \in \mathcal{I}(t) \bigcap \mathcal{B}} \check{\sigma}'^2_{\boldsymbol{z}_*|\mathcal{D}^{[i]}(t)} \leq \sum_{i \in \mathcal{I}(t) \bigcap \mathcal{B}} \max_{i \in \mathcal{V} \backslash \mathcal{B}} \left\{ \check{\sigma}^2_{\boldsymbol{z}_*|\mathcal{D}^{[i]}(t)} \right\}$$

$$\leq \sum_{i \in \mathcal{I}(t) \bigcap \mathcal{B}} \max_{i \in \mathcal{V} \backslash \mathcal{B}} \left\{ \sigma^2_f - \frac{\kappa(d^{[i]}(t))^2}{\sigma^2_f + (\sigma^{[i]}_e)^2} \right\} = |\mathcal{I}(t) \bigcap \mathcal{B}| \left( \sigma^2_f - \frac{\kappa(d^{\max}(t))^2}{\sigma^2_f + (\sigma^{\max}_e)^2} \right). \tag{23}$$

Therefore, combining (21) with (22) and (23), the upper bound of $\hat{\sigma}^2_{\boldsymbol{z}_*|\mathcal{D}(t)}$ is given as

$$\hat{\sigma}^2_{\boldsymbol{z}_*|\mathcal{D}(t)} \leq \sigma^2_f - \frac{\kappa(d^{\max}(t))^2}{\sigma^2_f + (\sigma^{\max}_e)^2}. \tag{24}$$

2) Lower bound. Similar to (21), we have

$$\sum_{i \in \mathcal{I}(t)} \check{\sigma}'^{-2}_{\boldsymbol{z}_*|\mathcal{D}^{[i]}(t)} = \sum_{i \in \mathcal{F}(t)} \check{\sigma}'^{-2}_{\boldsymbol{z}_*|\mathcal{D}^{[i]}(t)} + \sum_{i \in \mathcal{I}(t) \bigcap \mathcal{B}} \check{\sigma}'^{-2}_{\boldsymbol{z}_*|\mathcal{D}^{[i]}(t)}. \tag{25}$$

First, it holds that $\check{\sigma}'^{-2}_{\boldsymbol{z}_*|\mathcal{D}^{[i]}(t)} = \check{\sigma}^{-2}_{\boldsymbol{z}_*|\mathcal{D}^{[i]}(t)}$ for all $i \in \mathcal{F}(t)$. Theorem IV.3 in [29] gives $\check{\sigma}^2_{\boldsymbol{z}_*|\mathcal{D}^{[i]}(t)} \geq \frac{\sigma^2_f(\sigma^{[i]}_e)^2}{\sigma^2_f + (\sigma^{[i]}_e)^2}$ for all $\boldsymbol{z}_* \in \mathcal{Z}_*$. Then following the logic of deriving the above upper bound, we have

$$\sum_{i \in \mathcal{F}(t)} \check{\sigma}^{-2}_{\boldsymbol{z}_*|\mathcal{D}^{[i]}(t)} \leq \sum_{i \in \mathcal{F}(t)} \frac{\sigma^2_f + (\sigma^{[i]}_e)^2}{\sigma^2_f(\sigma^{[i]}_e)^2} \leq |\mathcal{F}(t)| \frac{\sigma^2_f + (\sigma^{\max}_e)^2}{\sigma^2_f(\sigma^{\min}_e)^2}. \tag{26}$$

Second, Lemma 2 implies the following inequality

$$\sum_{i \in \mathcal{I}(t) \bigcap \mathcal{B}} \check{\sigma}'^{-2}_{\boldsymbol{z}_*|\mathcal{D}^{[i]}(t)} \leq \sum_{i \in \mathcal{I}(t) \bigcap \mathcal{B}} \left( \min_{i \in \mathcal{V} \backslash \mathcal{B}} \left\{ \check{\sigma}^2_{\boldsymbol{z}_*|\mathcal{D}^{[i]}(t)} \right\} \right)^{-1} \leq \sum_{i \in \mathcal{I}(t) \bigcap \mathcal{B}} \left( \min_{i \in \mathcal{V} \backslash \mathcal{B}} \left\{ \frac{\sigma^2_f(\sigma^{[i]}_e)^2}{\sigma^2_f + (\sigma^{[i]}_e)^2} \right\} \right)^{-1}$$

$$\leq |\mathcal{I}(t) \bigcap \mathcal{B}| \frac{\sigma^2_f + (\sigma^{\max}_e)^2}{\sigma^2_f(\sigma^{\min}_e)^2}. \tag{27}$$

Therefore, combining (25) with (26) and (27) yields an upper bound of $\sum_{i \in \mathcal{I}(t)} \check{\sigma}'^{-2}_{\boldsymbol{z}_*|\mathcal{D}^{[i]}(t)}$, i.e., $\sum_{i \in \mathcal{I}(t)} \check{\sigma}'^{-2}_{\boldsymbol{z}_*|\mathcal{D}^{[i]}(t)} \leq |\mathcal{I}(t)| \frac{\sigma^2_f + (\sigma^{\max}_e)^2}{\sigma^2_f(\sigma^{\min}_e)^2}$. Recall that $\hat{\sigma}^2_{\boldsymbol{z}_*|\mathcal{D}(t)} = \frac{|\mathcal{I}(t)|}{\sum_{i \in \mathcal{I}(t)} \check{\sigma}'^{-2}_{\boldsymbol{z}_*|\mathcal{D}^{[i]}(t)}}$, then we have

$$\hat{\sigma}^2_{\boldsymbol{z}_*|\mathcal{D}(t)} \geq \frac{\sigma^2_f(\sigma^{\min}_e)^2}{\sigma^2_f + (\sigma^{\max}_e)^2}. \tag{28}$$

Thus, combining (24) and (28), the proof is complete.