# OpenReview forum: "Byzantine-tolerant federated Gaussian process regression for streaming data"
_NeurIPS.cc/2022/Conference — NeurIPS 2022 Accept_

### Official Review · Reviewer_3mHJ · 2022-07-09

**Rating:** 5
**Confidence:** 3
**Soundness:** 2 fair
**Presentation:** 2 fair
**Contribution:** 2 fair

**Summary:**

This paper studies consider Byzantine-tolerant federated learning for streaming data using Gaussian process regression (GPR). A cloud and a group of agents aim to collaboratively learn a latent function where some agents are subject to Byzantine attacks. The paper develops a Byzantine-tolerant federated GPR algorithm, which includes three modules: agent-based local GPR, cloud-based aggregated GPR and agent-based fused GPR.



**Questions:**

1. What are the technical challenges and contributions?
2. Is there a section of related work?
3. How to address the streaming data challenge, e.g., concept drifting?
4. How to simulate and test a data stream in experiments?

**Limitations:**

See the above questions.

**Strengths And Weaknesses:**

Strengths:

1. Develops a Byzantine-tolerant federated GPR algorithm. The algorithm includes three modules: agent-based local GPR, cloud-based aggregated GPR, and agent-based fused GPR.
2. The predictive mean of the cloud-based aggregated GPR is theoretically guaranteed to be restricted within a neighborhood of the target function provided that Byzantine agents are less than one-quarter of all the agents.
Experiments on a synthetic dataset and two real-world datasets are conducted to evaluate the proposed algorithm.

Weakness:
1. Technical contribution: the paper is a combination of existing methods on Byzantine attacks, federated GPR, and online learning, the technical challenges of such a combination, however, have not been identified, so the technical contribution of this work is not clear.
2. Related work: the paper does not include discussions with related work, there has been a sequence of works on Byzantine resilient secure federated learning, federated learning, online learning, and data streams. However, the difference between this work and theirs are not clearly stated.
3. Experiments: this paper targets data streams, but in experiments, there is no study in terms of data streams; for example, to simulate streaming data with concept drift. The measures should also include efficiency, as well as computation and memory costs.

---

> ### Author Response · Authors · 2022-08-02
> **Response to Reviewer 3mHJ**
>
> Thank you so much for your detailed and encouraging reviewing. The comments you give are valuable, and they can improve the quality of our paper. We address your comments as follows.
>
> Weaknesses:
>
> W1: Technical contribution: the paper is a combination of existing methods on Byzantine attacks, federated GPR, and online learning, the technical challenges of such a combination, however, have not been identified, so the technical contribution of this work is not clear.
>
> $Answer:$ Thank you so much for the comment on technical contribution. Although our work is in the intersection of Byzantine attack, GPR and federated learning, there are new and unique challenges to be addressed.
>
> First of all, most existing literature on Byzantine attacks mostly focus on deep neural networks as the machine learning model. Please refer to references [C1], [C2] and [C3]. The technical challenges typically rely on the convergence of SGD under Byzantine attacks. In our paper, we focus on Byzantine attacks on the predictions of GPR as the machine learning model. The technique challenge lies on the convergence of GPR under Byzantine attacks.
>
> Second, to the best of our knowledge, this is the first paper to analyze the Byzantine tolerance of federated GPR models. We theoretically guarantee that the global predictive mean in the cloud converges to a neighborhood of the ground truth regardless of Byzantine attacks.
>
> W2: Related work: the paper does not include discussions with related work, there has been a sequence of works on Byzantine resilient secure federated learning, federated learning, online learning, and data streams. However, the difference between this work and theirs are not clearly stated.
>
> $Answer:$ Thank you so much for the comment on related work. In fact, our introduction includes the related work, which is explicitly described in the second paragraph. In our revised paper, we highlight and group this section under "related work" to improve the clarity.
>
> W3: Experiments: this paper targets data streams, but in experiments, there is no study in terms of data streams; for example, to simulate streaming data with concept drift. The measures should also include efficiency, as well as computation and memory costs.
>
> $Answer:$ Thank you for the valuable comment on our experiments. Although the data pool is static, we simulate data stream by feeding the agents with data sequentially. Specifically, at the initial time, $\mathcal{D}^{[i]}(0)=\emptyset$. As time arises, each agent $i\in\mathcal{V}$ collects data online and updates its local dataset by $\mathcal{D}^{[i]}(t)=\mathcal{D}^{[i]}(t-1)\bigcup(\mathcal{Z}^{[i]}(t),\boldsymbol{y}^{[i]}(t))$. For each test input $z_*\in\mathcal{Z}_{\star}$, we sequentially execute the agent-based local GPR, the cloud-based aggregated GPR and the agent-based fused GPR.
>
> This is a standard way to evaluate online learning algorithms with streaming data. Please refer to [C4],[C5] and references therein. Concept drift in supervised learning indicates that the underlying distribution of the data is changing, and consequently the predictions might become less accurate as the time passes [C6], [C7]. However, our paper focuses on Byzantine tolerance only. Therefore in the experiments, we only study the performance of our algorithm under Byzantine attacks. Studying how our algorithm performs under concept drift is an interesting future work.
>
> Questions:
>
> Q1: What are the technical challenges and contributions?
>
> $Answer:$ Thank you for your question. First, we develop a novel trimming rule by removing a fraction of extreme values on local predictive means and variances such that product-of-experts (PoE) computation is secured. Please refer to Lemma 2, which guarantees the robustness of the trimming rule.
>
> Second, in the attack-free scenario, the convergence of prediction error for the cloud-based aggregated GPR is guaranteed in Corollary 1. This is a new theoretical result. Further, when we consider Byzantine agents in the network, problem becomes harder. This is because the Byzantine agents can make the learning performance of existing PoE arbitrarily bad. Even so, Theorem 1 in our paper guarantees the robustness of the Byzantine-tolerant PoE.
>
> Q2: Is there a section of related work?
>
> $Answer:$ Thank you for your question. In our revised version, we add the key words ``Related work'' before this paragraph to make it clearer.

---

> ### Author Response · Authors · 2022-08-02
> **Response to Reviewer 3mHJ (Cont'd)**
>
>
> Q3: How to address the streaming data challenge, e.g., concept drifting?
>
> $Answer:$ Many thanks for the question. There are many challanges in streaming data, including Byzantine attacks, concept drift and variety of formats.
>
> This paper considers Byzantine attacks only. In fact, to deal with concept drift, one idea is to combine with existing work [C8]. Specifically, we can develop a method to detect the changes of the probability distribution of samples, and control the error rate of the data.
>
> Q4: How to simulate and test a data stream in experiments?
>
> $Answer:$ Please refer to our answers to W3. Thank you so much.
>
> References
>
> [C1] D. Yin, Y. Chen, R. Kannan, and P. Bartlett, “Byzantine-robust distributed learning: Towards
> optimal statistical rates,” in International Conference on Machine Learning, pp. 5650–5659,
> PMLR, 2018.
>
> [C2] P. Blanchard, E. M. El Mhamdi, R. Guerraoui, and J. Stainer, “Machine learning with adversaries:
> Byzantine tolerant gradient descent,” in Advances in Neural Information Processing Systems
> (I. Guyon, U. V. Luxburg, S. Bengio, H. Wallach, R. Fergus, S. Vishwanathan, and R. Garnett,
> eds.), vol. 30, Curran Associates, Inc., 2017.
>
> [C3] Y. Chen, L. Su, and J. Xu, “Distributed statistical machine learning in adversarial settings:
> Byzantine gradient descent,” Proceedings of the ACMonMeasurement and Analysis of Computing
> Systems, vol. 1, no. 2, pp. 1–25, 2017.
>
> [C4] E. Beyazit, J. Alagurajah, and X. Wu, “Online learning from data streams with varying feature spaces,” in AAAI Conference on Artificial Intelligence, 2019.
>
> [C5] B. J. Hou, L. Zhang, and Z. H. Zhou, “Learning with
> feature evolvable streams,” in Advances in Neural Information Processing Systems, 2017.
>
> [C6] S. Wang, L. L. Minku, D. Ghezzi, D. Caltabiano, P. Tino, and X. Yao,
> “Concept drift detection for online class imbalance learning,” in Neural
> Networks (IJCNN), The 2013 International Joint Conference on. IEEE,
> 2013, pp. 1–10.
>
> [C7] R. Klinkenberg and T. Joachims, “Detecting concept drift with support 	vector machines,” in Proceedings of the Seventeenth International Conference on Machine Learning. Morgan Kaufmann Publishers Inc.,
> 2000, pp. 487–494.
>
> [C8] J. Gama, P. Medas, G. Castillo, and P. Rodrigues, “Learning with drift detection,” in Advances in Artificial Intelligence–SBIA. Springer, pp. 286–295, 2004.

---

### Official Review · Reviewer_vGEs · 2022-07-11

**Rating:** 5
**Confidence:** 4
**Ethics Flag:** Yes
**Soundness:** 2 fair
**Presentation:** 2 fair
**Contribution:** 2 fair

**Summary:**

The paper explains the Byzantine-tolerated local approximation Gaussian process (GP) model for streaming data. In the GP framework, distributed learning scenario is used to scale the costly GP to big data. Besides, this local approximation method is used in federated learning when security requirements should be satisfied. The paper proposes a solution to divide the agent set into Benign and Byzantine agents sets. By specifying the Byzantine agents, the model excludes them and uses the benign agents to reduce the negative effects of the attack. The authors evaluate the proposes model on synthetic and real world data sets to show the capacity of their solution.

**Questions:**

Please check the points mentioned in "Strengths And Weaknesses.". Indeed, it has been confirmed that disjoint partitioning captures the local features more accurately and outperforms random partitioning (see reference [23] in the paper); why did the authors use random partitioning in the experiments?


**Limitations:**

Yes.

**Strengths And Weaknesses:**

Strengths:

The paper correctly defines the problem. It explains the model, restrictions, related assumptions, and the main problem by addressing all related issues.

Weaknesses:
The paper has some shortcomings, and I will mention them here:

1- Agent-based local GPR:

The size of the local input data in an agent is not determined. When time rises (i.e., t goes to infinity), how the model handles the data assigned to each agent? In streaming data, the size of the input data increases continuously. Since the local models use the nearest neighborhood GP regression, how many streaming inputs should be selected for NNGP?
The size of the data inputs raises, so the model needs a clearly defined strategy that determines the local partitions' scope, the number of neighborhood points in NNGP, and related complexity. The computational cost of a GP has a cubic dependency on the sample size (in the local approximations, cubically depends on the subsample sizes). Since the sample size increases when time t raises, the model needs a predefined mechanism to deal with complexity. Moreover, the complexity O(t) mentioned in line 146 only shows the search cost in NN and not the cost of the training of the NNGP.


2- Cloud-based aggregated GPR

The paper's main contribution is this part, where they proposed a solution to find Byzantine agents. The paper considers the agents with the smallest and largest local means and variances at each test point as potential Byzantine agents. It is not supported by a theoretical analysis in the paper and is a heuristic trick. Consider a simple case that all Byzantine agents reported small values as local predictions while the higher mean values are more proper. In this case, steps 4 to 7 in Algorithm 3 exclude a small number of Byzantine agents, while some high-quality agents are also excluded.

There are some related selection methods in the literature. For instance, the Gaussian graphical model can select the benign agents using the precision matrix between agents' predictions. The agent selection method in (https://arxiv.org/abs/2102.01496.pdf)  uses the interactions between agents in the related graphical model, determines those agents whose predictions are not close to the other agents' predictions, and excludes them for the final aggregation. Indeed, the ensemble technique in (https://arxiv.org/pdf/2010.08873.pdf) can divide the agents set into two clusters, Byzantine and benign agents, using spectral clustering and the precision matrix of the local predictions. Besides, the importance of an agent in the original GPoE (https://arxiv.org/pdf/1410.7827.pdf) and robust Bayesian committee machine (https://arxiv.org/pdf/1502.02843.pdf) can also help to select benign agents. They proposed the difference in the differential entropy between the prior and the local posterior distribution.

3-  Byzantine-tolerant PoE

The generalized product of experts (GPoE) with equal weights (Deisenroth et al. 2015) can provide conservative predictions. This model averaging approach uses the agents' weights to improve the overconfident predictions of traditional PoE. However, its prediction quality is generally lower than the Bayesian committee machine family, mainly when disjoint partitioning is used. It would be better if the authors considered the other newer aggregation methods.

4- Agent-based fused GPR

This part is somewhat unclear. When the local predictions have been combined using PoE, what is the reason for this step? It should be explained that can it change the final aggregation? The model compares the variance of the GPoE and local variances.
The reason for comparison is unclear, and also, the paper does not describe why this comparison and related replacement (steps 3 and 5 in Algorithm 4) are essential?

---

> ### Author Response · Authors · 2022-08-02
> **Response to Reviewer vGEs**
>
> Thank you so much for reviewing our paper. We are grateful and indebted for the time invested to evaluate our paper, and for the suggestions to make our paper a better and stronger contribution.
>
> Weaknesses:
>
> W1 Agent-based local GPR:
>
> W1-1: The size of the local input data in an agent is not determined. When time rises (i.e., t goes to infinity), how the model handles the data assigned to each agent?
>
> $Answer:$ Thank you so much for your comment on the size of local datasets. In fact, our algorithm of agent-based local GPR does not assign the data to each agent. Instead, the agents collect data independently. As in Line 6 of Algorithm 1, each agent $i\in\mathcal{V}$ collects data online and updates its local dataset by $\mathcal{D}^{[i]}(t)=\mathcal{D}^{[i]}(t-1)\bigcup(\mathcal{Z}^{[i]}(t),y^{[i]}(t))$. We admit that better learning performance requires an infinite amount training data in theory, but it is impractical in the applications. In fact, there is a tradeoff between theory and engineering. Full GPR uses all training data to make better predictions, and the corresponding computation complexity is $\mathcal{O}(t^3)$. Hence to reduce the computation complexity in practice, we have to limit the size of the local input data in an agent. We can either adopt a simple practice such as stop collecting data when the size of training data reaches certain threshold or use existing work [B1] to only keep a fix number of most informative data points.
>
> Our algorithm only is associated with one training data by using projection, and covariance between two training inputs is $0$. Although our algorithm has learning performance loss, the computation complexity is reduced to $\mathcal{O}(t)$, which is practical in the applications.
>
> W1-2: In streaming data, the size of the input data increases continuously. Since the local models use the nearest neighborhood GP regression, how many streaming inputs should be selected for NNGP?
>
> $Answer:$ Thank you for your question. In this paper, instead of feeding the whole training dataset to GPR, the agent-based local GPR only uses the nearest input denoted by $z_*^{[i]}(t)\in(z_{\star},\mathcal{Z})$ and its corresponding output $y_{z_*^{[i]}(t)}^{[i]}$ to compute the local predictions. Therefore, we only choose one training point by projection to compute the local predictions.
>
> W1-3: The size of the data inputs raises, so the model needs a clearly defined strategy that determines the local partitions' scope, the number of neighborhood points in NNGP, and related complexity. The computational cost of a GP has a cubic dependency on the sample size (in the local approximations, cubically depends on the subsample sizes). Since the sample size increases when time t raises, the model needs a predefined mechanism to deal with complexity. Moreover, the complexity O(t) mentioned in line 146 only shows the search cost in NN and not the cost of the training of the NNGP.
>
> $Answer:$ We very appreciate the suggestions to train a NNGP on the determination of local training data size and complexity reduction. We agree that the computation complexity of the hyperparameter tuning for NNGP is $\mathcal{O}(t^3)$. In fact, the computation complexity of the hyperparameter tuning can be reduced to $\mathcal{O}(1)$ by using recursive hyperparameter tuning (please refer to reference [B2]). We briefly mention this in the revised version in the subsection of the agent-based local GPR. Furthermore, this paper considers the computation complexity in prediction, which is also $\mathcal{O}(t^3)$ for full GPR. By using NNGP, this complexity is reduced to $\mathcal{O}(t)$.

---

> > ### Comment · Reviewer_vGEs · 2022-08-04
> > **Thanks for the clarifications of computational costs**
> >
> > Thanks for the clarifications of computational costs in the proposed model. The training cost of a full GP is $\mathcal{O}(t^3)$ while the prediction cost is  $\mathcal{O}(t^2)$ due to the vector-matrix operation in Equation (3) (it would be better to modify it in the paper).
> > I am not sure whether training a local GP with only one training point makes sense or not. However, the authors' answer, " In this paper, instead of feeding the whole training dataset to GPR, the agent-based local GPR only uses the nearest input denoted," contradicts the procedure in the experiments. For instance, see Lines 279 and 280, "In the experiment, we partition 40 000 training data
> >  into n = 40 disjoint groups, and assign each group to an agent".

---

> > > ### Author Response · Authors · 2022-08-07
> > > **Response to Reviewer vGEs**
> > >
> > > $Answer:$ Thank you so much for the suggestion, and we modify it in our revised paper. In this paper, we train GPR using standard method and using all the local data. But the local GPR makes prediction for each test inputs using only one data point that is nearest to the test point. This reduces the computation complexity for prediction to the complexity of nearest neighbor search, which is $\mathcal{O}(t)$ in the worst case.
> > >
> > > Similarly in the experiment, we use the partitioned data to train the GPR. But after training, for prediction, we only use one data point for each test input. Thank you for pointing this out and we also refine the description of the procedure in our revised paper.

---

> ### Author Response · Authors · 2022-08-02
> **Response to Reviewer vGEs (Cont'd)**
>
> W2 Cloud-based aggregated GPR:
>
> W2-1: The paper's main contribution is this part, where they proposed a solution to find Byzantine agents. The paper considers the agents with the smallest and largest local means and variances at each test point as potential Byzantine agents. It is not supported by a theoretical analysis in the paper and is a heuristic trick. Consider a simple case that all Byzantine agents reported small values as local predictions while the higher mean values are more proper. In this case, steps 4 to 7 in Algorithm 3 exclude a small number of Byzantine agents, while some high-quality agents are also excluded.
>
> $Answer:$ We really appreciate the comment on the cloud-based aggregated GPR. We agree that our trimming rule could exclude some high-quality agents. However, Lemma 2 theoretically guarantees the robustness of the trimming rule, which can further secure the computation of PoE with the remaining agents' local predictions. Please note that Byzantine agents can send arbitrary messages to the cloud. No one can theoretically guarantee that the proposed agent selection algorithm is perfect [B1]-[B5]. If the algorithm misses any malicious agent, the learning performance of existing PoE could be arbitrarily bad. In contrast, the learning performance of our Byzantine-tolerant GPR is bounded, and depends on some parameters, e.g., the number of Byzantine agents. Please refer to Theorem 1. The first two experiments in our paper are conducted to demonstrate this result. Therefore, our design is not heuristic.
>
> W2-2: There are some related selection methods in the literature. For instance, the Gaussian graphical model can select the benign agents using the precision matrix between agents' predictions. The agent selection method in (https://arxiv.org/abs/2102.01496.pdf) uses the interactions between agents in the related graphical model, determines those agents whose predictions are not close to the other agents' predictions, and excludes them for the final aggregation. Indeed, the ensemble technique in (https://arxiv.org/pdf/2010.08873.pdf) can divide the agents set into two clusters, Byzantine and benign agents, using spectral clustering and the precision matrix of the local predictions. Besides, the importance of an agent in the original GPoE (https://arxiv.org/pdf/1410.7827.pdf) and robust Bayesian committee machine (https://arxiv.org/pdf/1502.02843.pdf) can also help to select benign agents. They proposed the difference in the differential entropy between the prior and the local posterior distribution.
>
> $Answer:$ Thank you so much for your good suggestion on selection methods. We agree that these papers provide different methods on agents selection. However, existing agent selection methods require information, which is not available for the cloud.
>
> Specifically, on the one hand, the first two methods utilize interactions between agents in a graph. However, in our network model or the setup of federated learning, there is no edge between any two agents. In particular, in terms of paper (https://arxiv.org/abs/2102.01496.pdf), the full experts set is divided into two subsets, important and unimportant experts. The importance of an expert is measured according to interactions with other experts. By definition of importance and unimportance (see its Definition 2 on page 4), $M_{\alpha}$ is the set of important experts, if it contains $\alpha\times100$ of the most connected experts in graph $\mathcal{G}$, where $\alpha\in[0,1]$. Its complement, $\tilde{{M}}_{\alpha}$ contains the remaining unimportant experts. There is no connection between agents in our problem, and we cannot measure the importance and unimportance. Similar to (https://arxiv.org/pdf/2010.08873.pdf). Therefore, such strategies cannot be used in our problem.
>
> On the other hand, in terms of gPoE (https://arxiv.org/pdf/1410.7827.pdf) and rBCM (https://arxiv.org/pdf/1502.02843.pdf), at a given point $x$, for each agent $i\in\mathcal{V}$, the weighting term $\alpha_i(x)$ is introduced to compute the global mean and variance. This term can control the influence of individual experts and help to partition groups. Note that typical choice of $\alpha_i(x)$ is the change in entropy from prior to posterior at point $x$, or the difference between the prior and posterior variance. First, we do not assume that the cloud has any prior local information for each agent $i\in\mathcal{V}$, e.g., entropy and variance, in our paper. Second, even though the cloud is able to obtain the potentially corrupted $\alpha_i(x)$ (since the local posterior predictions the cloud receives could be corrupted by Byzantine attacks), it is arbitrary as well. We also need to find an algorithm to construct an agent set where some agents' local information can be used for aggregation. It is not straightforward. Thanks again for good suggestions.

---

> > ### Comment · Reviewer_vGEs · 2022-08-04
> > **Comparison with other agent selection methods**
> >
> >  Thanks for your response. I need to first talk about a misunderstanding about the suggested methods.
> > In the experiments, the proposed method has been compared with attacked PoE and attacked-free PoE, which is insufficient. Therefore, using the suggested baselines (e.g., graph-based models or weighting-based models) in the experiments can demonstrate the efficacy of the proposed method.
> >
> > However, I want to add some details about the authors' responses.
> > 1- The suggested graph-based methods convert the set of agents into a graphical model. However, instead of the initial data, they use the means of the local predictive distributions. When the local predictions of an agent are not close to those of the other agents, the graphical model defines this agent as an unimportant expert and excludes it from final aggregation. The proposed method in this paper also determines an agent as Byzantine if its moments of local predictive distribution (mean and variance) are far from those of the other agents (see Equations 6 and 7). Therefore, the strategies to determine a Byzantine or unimportant agent and inputs in both papers are the same. The interactions between the agents in the graphical models are not a significant issue here. However, the graph-based papers considered independent agents and showed the methods also work when there is no interaction.
> > On the other hand, the main disadvantage of the graph-based models is that they assign a fixed set of agents to all new data points and are not flexible in capturing the specific behavior of the new test points. While the proposed method in the paper is a flexible pointwise method, and the selection method is repeated for further test points. The weighting-based models can also provide different baselines for comparison.
> >
> > 2- We can accept the selection trick here is not a complete heuristic method (due to the discussion in Section 3.4). However, it only considers too simple attacks. It assumes that a Byzantine agent always sent a simple arbitrary message far from the other messages. It can be easily discovered by removing the large and small values in Equation 6. Thus, the algorithm does not explain how a Byzantine agent can be separated from a weak benign expert (an agent which provides low-quality predictions). As an example of the model limitation, consider the example in the review when all Byzantine agents send small values as arbitrary messages, and the model can not detect all of them. Instead, it removes some benign agents that provide the largest local predictive means.

---

> > > ### Author Response · Authors · 2022-08-07
> > > **Response to Reviewer vGEs**
> > >
> > > Thank you so much for reviewing our paper, and the comments you give are valuable, which can improve the quality of our paper.
> > >
> > > Comment 1: In the experiments, the proposed method has been compared with attacked PoE and attacked-free PoE, which is insufficient. Therefore, using the suggested baselines (e.g., graph-based models or weighting-based models) in the experiments can demonstrate the efficacy of the proposed method.
> > >
> > > $Answer:$ Thank you so much for your comment on the experiments.
> > >
> > > The suggested comparisons are interesting and will make the experiments stronger. However, our current experiments are sufficient to demonstrate the effectiveness of our algorithm. The reasons are twofold.
> > >
> > > First of all, the objective of our paper is to design an algorithm which can enable the agents and the cloud to correctly learn the function $\eta$ without requiring the agents to share local streaming data $({{z}^{[i]}(t), y^{[i]}(t)})$ despite some agents are subject to Byzantine attacks. We do not aim to distinguish Byzantine agents and benign agents. Hence the experiments are conducted to show the proposed algorithm is able to perform correct predictions regardless of Byzantine attacks, instead of showing our agent selection strategy is superior to others.
> > >
> > > Second, our experiment setup is consistent with the literature on Byzantine-tolerant machine learning [D1]-[D4]. The experiments of all these references including our paper focus on the correct predictions regardless of Byzantine attacks and how the number of Byzantine agents affects learning error (convergence rate). For example, paper [D1] uses MNIST dataset to show the effectiveness of the proposed agent selection methods. This paper trains a multi-class logistic regression model and a convolutional neural network (CNN) using distributed gradient descent, and for each model, the paper compares the learning accuracy in different settings: attack-free standard aggregation rule, attacked standard aggregation rule, and Byzantine-resilient aggregation rule. Paper [D2] considers the task of spam filtering. The learning model is a multi-layer perceptron (MLP) with two hidden layers. The paper compares learning accuracy in different settings: attack-free standard aggregation rule, attacked standard aggregation rule, and Byzantine-resilient aggregation rule, and analyzes the learning error with different number of Byzantine agents. Both papers [D1] and [D2] conclude that standard aggregation rule cannot tolerate Byzantine agents, while their proposed aggregation rule can. Our paper follows the same experiment setup to demonstrate that our proposed aggregation rule can tolerate Byzantine agents and the performance of the Byzantine-tolerant PoE increases as the ratio of Byzantine agents decreases.
> > >
> > > Lastly, we would like to mention that the major contribution of the paper lies in the theoretical guarantees (Theorem 1 and Theorem 2).

---

> > > ### Author Response · Authors · 2022-08-07
> > > **Response to Reviewer vGEs (Cont'd)**
> > >
> > > Comment 2: We can accept the selection trick here is not a complete heuristic method (due to the discussion in Section 3.4). However, it only considers too simple attacks. It assumes that a Byzantine agent always sent a simple arbitrary message far from the other messages. It can be easily discovered by removing the large and small values in Equation 6. Thus, the algorithm does not explain how a Byzantine agent can be separated from a weak benign expert (an agent which provides low-quality predictions). As an example of the model limitation, consider the example in the review when all Byzantine agents send small values as arbitrary messages, and the model can not detect all of them. Instead, it removes some benign agents that provide the largest local predictive means.
> > >
> > > $Answer:$ Thank you so much for the comment on our trimming rule. As we clarify, we do not aim to separate Byzantine agents from benign agents. This is completely different from the agent selection methods in [D5]-[D8]. Papers [D1]-[D4] including our paper on Byzantine-tolerant machine learning dedicate to guaranteeing the robustness of the proposed algorithms without correctly identifying Byzantine agents, and their theories and ours hold even when malicious agents are included and benign agents are excluded.
> > >
> > > We take a toy example to illustrate. Assume that there are 5 agents in the network. Agent 1: $\mu_1=1$; Agent 2: $\mu_2=2$; Agent 3: $\mu_3=3$; Agent 4: $\mu_4 = 4$; Agent 5: $\mu_5=5$. The ground truth average is $3$. We assume that agent 3 is a Byzantine agent. We consider two cases as follows.
> > >
> > > Case 1): The Byzantine agent changes $\mu_3=3$ to an extreme value ($\mu_3'>5$ or $\mu_3'<1$). Since we assume that the cloud is aware of the number of Byzantine agents ($|\mathcal{B}|=1$), agent 3 is removed by our trimming rule (removing a largest value and a smallest value). The computed average is $3$, which is the ground truth average.
> > >
> > > Case 2): The Byzantine agent changes $\mu_3=3$ to $\mu_3'\in(1,5)$. By applying our trimming rule, the values of agents 2, 3 and 4 are kept, and the computed average belongs to $(\frac{7}{3},\frac{11}{3})$, which is around the ground truth average.
> > >
> > > The above example provides the following insights. First, when the cloud receives $\mu_1$, $\mu_2$, $\mu_3'$, $\mu_4 $ and $\mu_5$, it cannot identify which one is the Byzantine agent; Second, even though some benign agents are removed and Byzantine agents are included, the computed averages are close to the ground truth average. Third, it is not always optimal for the Byzantine agent to choose extreme values. In case 1, the Byzantine agent chooses an extreme value but the ground truth average can be obtained.
> > >
> > > Therefore, our trimming rule is applicable to Byzantine agents with $arbitrary$ $messages$. It is interesting to adopt the suggested agent selection methods in our algorithm and theoretically analyze their robustness. We leave it as a future work.
> > >
> > > References
> > >
> > > [D1] D. Yin, Y. Chen, R. Kannan, and P. Bartlett, “Byzantine-robust distributed learning: Towards
> > > optimal statistical rates,” in Proceedings of the International Conference on Machine Learning,
> > > pp. 5650–5659, 2018.
> > >
> > > [D2] P. Blanchard, E. M. El Mhamdi, R. Guerraoui, and J. Stainer, “Machine learning with adversaries:
> > > Byzantine tolerant gradient descent,” in Proceedings of International Conference on Neural
> > > Information Processing Systems, pp. 118–128, 2017.
> > >
> > > [D3] D. Data and S. Diggavi, “Byzantine-resilient high-dimensional SGD with local iterations on
> > > heterogeneous data,” in Proceedings of the International Conference on Machine Learning,
> > > pp. 2478–2488, 2021.
> > >
> > > [D4] L. Li, W. Xu, T. Chen, G. B. Giannakis, and Q. Ling, “RSA: Byzantine robust stochastic aggregation methods for distributed learning from
> > > heterogeneous datasets,” in Proceedings of the AAAI Conference on Artificial Intelligence, pp. 1544–1551, 2019.
> > >
> > > [D5] H. Jalali, M. Pawelczyk, and G. Kasneci, ``Gaussian experts selection using graphical models,'' https://arxiv.org/abs/2102.01496, 2021.
> > >
> > > [D6] H. Jalali and G. Kasneci, ``Aggregating Dependent Gaussian experts in local approximation,'' https://arxiv.org/abs/2010.08873, 2020.
> > >
> > > [D7] Y. Cao and D. J. Fleet, ``Generalized product of experts for automatic and principled fusion of Gaussian process predictions,'' https://arxiv.org/abs/1410.7827, 2014.
> > >
> > > [D8] M. P. Deisenroth and J. W. Ng, ``Distributed Gaussian processes,'' in Proceedings of the 32nd International Conference on Machine Learning, 2015.

---

> ### Author Response · Authors · 2022-08-02
> **Response to Reviewer vGEs (Cont'd)**
>
>
> W3 Byzantine-tolerant PoE:
>
> W3: The generalized product of experts (GPoE) with equal weights (Deisenroth et al. 2015) can provide conservative predictions. This model averaging approach uses the agents' weights to improve the overconfident predictions of traditional PoE. However, its prediction quality is generally lower than the Bayesian committee machine family, mainly when disjoint partitioning is used. It would be better if the authors considered the other newer aggregation methods.
>
> $Answer:$ Many thanks for the valuable suggestion on the aggregation rules in the could. We agree that gPoE can be more conservative in predictions. However, our current theoretical results only support gPoE, and can be rigorously proved for solving the problem of Byzantine-tolerant federated GPR. Generalizing our theoretical results to the BCM family requires additional analysis and is part of our ongoing work.
>
> W4 Agent-based fused GPR:
>
> W4-1: This part is somewhat unclear. When the local predictions have been combined using PoE, what is the reason for this step?
>
> $Answer:$ Thank you for the comment on the agent-based fused GPR. Notice that cloud-based aggregated GPR makes predictions using data from all the agents while agent-based local GPR makes predictions using local data only. Therefore, the cloud-based GPR tends to have better performances compared to agent-based local GPR. Since eventually it is the agents who make predictions, fused GPR allows the agents to utilize the predictions from the cloud to potentially enhance predictions by leveraging communication.
>
> W4-2: It should be explained that can it change the final aggregation?
>
> $Answer:$ We appreciate the good suggestion. Our algorithm is one-round. The refined predictions in the agent-based local GPR will not be transmitted to the cloud, hence it cannot change the final aggregation. As you suggest, we add the explanation in our revised version.
>
> W4-3: The model compares the variance of the GPoE and local variances. The reason for comparison is unclear, and also, the paper does not describe why this comparison and related replacement (steps 3 and 5 in Algorithm 4) are essential?
>
> $Answer:$ Thank you so much for the comment on the fused GPR. The design relies on the intuition that the predictive variance reflects the uncertainties in the predictions. Predictions with lower variance, corresponding to lower predictive uncertainties, usually implies higher prediction accuracy. Therefore, fused GPR replaces the predictions of the agent-based local GPR with those of the cloud aggregate GPR if the variance from the cloud is lower than that from the local GPR.
>
> Questions:
>
> Q1: Indeed, it has been confirmed that disjoint partitioning captures the local features more accurately and outperforms random partitioning (see reference [23] in the paper); why did the authors use random partitioning in the experiments?
>
> $Answer:$ We appreciate your careful reading. In fact, in our experiments, we partition the dataset into disjoint groups, and please refer to the synthetic dataset on page vii. We also partition the training dataset into disjoint groups. We find that the word "randomly" there can be confusing. Therefore, we revise the description in our revised paper. Thank you so much for pointing this out.
>
> References
>
> [B1] Lawrence, Neil, Matthias Seeger, and Ralf Herbrich. ``Fast sparse Gaussian process methods: The informative vector machine.'' Advances in neural information processing systems 15, 2002.
>
> [B2] Huber, Marco F. ``Recursive Gaussian process: On-line regression and learning.'' Pattern Recognition Letters 45, 85-91, 2004
>
> [B3] P. Blanchard, E. M. El Mhamdi, R. Guerraoui, and J. Stainer, “Machine learning with adversaries:
> Byzantine tolerant gradient descent,” in Proceedings of International Conference on Neural
> Information Processing Systems, pp. 118–128, 2017.
>
> [B4] J. So, B. Güler, and A. S. Avestimehr, “Byzantine-resilient secure federated learning,” IEEE
> Journal on Selected Areas in Communications, vol. 39, no. 7, pp. 2168–2181, 2020.
>
> [B5] D. Yin, Y. Chen, R. Kannan, and P. Bartlett, “Byzantine-robust distributed learning: Towards
> optimal statistical rates,” in Proceedings of the International Conference on Machine Learning,
> pp. 5650–5659, 2018.
>
> [B6] D. Data and S. Diggavi, “Byzantine-resilient high-dimensional SGD with local iterations on
> heterogeneous data,” in Proceedings of the International Conference on Machine Learning,
> pp. 2478–2488, 2021.
>
> [B7] Y. Chen, L. Su and J. Xu, ``Distributed statistical machine learning in adversarial settings: Byzantine gradient descent,'' Proceedings of the ACM on Measurement and Analysis of Computing Systems, vol. 1, no. 2, pp. 1–25, 2017.

---

> > ### Comment · Reviewer_vGEs · 2022-08-04
> > **About the authors' response to the agent-based fused GPR algorithm**
> >
> > Thank you for clarifying the points, especially in W4. The cloud-based aggregated GPR is the main output of the model, and it has been reported in the experiments. Thus, the proposed model does not use the results from the agent-based fused GPR algorithm. Besides, in the range of the data, GPoE overestimates the variance (predictions are conservative), especially with an increasing number of GP experts (see: Deisenroth et al. 2015, distributed Gaussian processes). It seems the comparison between variances in step 2 and step 5 does not lead to many changes.

---

> > > ### Author Response · Authors · 2022-08-07
> > > **Response to Reviewer vGEs**
> > >
> > > $Answer:$ We really appreciate the comment on the aggregation rule in the cloud.
> > >
> > > First of all, we would like to highlight that this paper is about Byzantine resiliency and needs to demonstrate the trimming rule is effective. The Byzantine resiliency is supported by our theory and experiments when gPoE is used.  It is not trivial to obtain theoretical guarantees when other PoE or rBCM is used. We leave the extension as a future work.
> > >
> > > Table 1 Prediction performance comparisons between local GPR and fused GPR on agent 1 and 6.
> > > _____________________________________________________
> > > Training data size   |    6000    |    10000   |    40000    |    150000
> > > _____________________________________________________
> > > MSE/i=1/local $\quad$ | 0.0721 | 0.0681 | 0.0585 | 0.0527
> > > _____________________________________________________
> > > MSE/i=1/fused $\quad$ |0.0096|0.0681|0.0094|0.0537
> > > _____________________________________________________
> > > MSE/i=6/local $\quad$ |0.0727|0.0566|0.0564|0.0555
> > > _____________________________________________________
> > > MSE/i=6/fused $\quad$|0.0727|0.0086|0.0564|0.0076
> > > _____________________________________________________
> > >
> > > Second, we conduct additional experiments to demonstrate the improvements by applying our gPoE and fusion algorithm to the synthetic dataset. For a test point, the local variances $\check{\sigma}'^{2}_{z*|\mathcal{D}(t)}$ lie within $[0.3660,0.6228]$, and the global variance is $\hat{\sigma}'^2=0.4678$. The prediction performance comparisons between the agent-based local GPR and fused GPR on agent $1$ and $6$ are listed in Table 1. It shows that the learning performance of the agent-based fused GPR is better than that of the agent-based local GPR. Note that the order of the prediction errors for the agent-based fused GPR can decrease from $10^{-2}$ to $10^{-3}$. This demonstrates that the comparison between variances in step 2 and step 5 in our proposed fusion algorithm is able to improve the learning accuracy, and our algorithm is effective. We will incorporate the new experimental results and justifications in our revised paper.

---

### Official Review · Reviewer_dgxv · 2022-07-12

**Rating:** 6
**Confidence:** 5
**Soundness:** 2 fair
**Presentation:** 3 good
**Contribution:** 2 fair

**Summary:**

The authors consider applying Byzantine-tolerant federated learning for online GPR and proposed a new algorithm with various modules for IoT applications. Theoretical supports on the proposed algorithms are also provided.


**Questions:**

1. Do assumptions 2 and 3 hold even for the SE kernel? This is not justified in the paper.
2. Are the bounds tight in Theorem 1 and 2?

**Ethics Review Area:**

["I don’t know"]

**Limitations:**

1. Insufficient algorithm comparison.
2. Insufficient experimental evaluations.

**Strengths And Weaknesses:**

Strengthes:
1. A novel Byzantine-tolerant federated GPR algorithm for online streaming data.
2. Theoretical performance analyses on the posterior mean and variance subject to attacks.

Weaknesses:
1. The main ideas and techniques rely on reference [24] to certain extent.
2. Attack is perhaps too simple.
3. Comparison with other methods is insufficient, for instance,
(a) Gaussian process with differential privacy, https://arxiv.org/abs/2106.00474
(b) Reference [23]
4. The following reference also discussed about the malicious protection and privacy issue of distributed GP for IoT applications, which may be mentioned in the manuscript:
https://ieeexplore.ieee.org/abstract/document/9250516

---

> ### Author Response · Authors · 2022-08-02
> **Response to Reviewer dgxv**
>
> Thank you so much for reviewing our paper, and the comments you give are valuable, which can improve the quality of our paper.
>
> Weaknesses:
>
> W1: The main ideas and techniques rely on reference [24] to certain extent.
>
> $Answer:$ Thanks so much for the comment on the main ideas and techniques of our paper. We agree that both paper [24] and our paper study Byzantine resilience of federated learning. However, they are different in terms of problem formulation, machine learning model, algorithm and theoretical guarantees. Specifically, paper [24] considers batch learning and adopts deep neural networks as the machine learning model, while our paper focuses on online learning and uses Gaussian process regression (GPR) as the learning model. In terms of algorithm of Byzantine tolerance, paper [24] designs a Byzantine robust aggregation algorithm for stochastic gradient descent to solve an optimization problem, while our paper designs a Byzantine-tolerant product-of-experts (PoE) algorithm to aggregate posterior distributions from local agents. Furthermore, in paper [24], Byzantine agents only send one variable, the malicious gradients, to the cloud, while in our paper, Byzantine agents could send two variables, malicious predictive means and variances. Since the global predictive mean depends on these two coupling variables using aggregation rules, e.g., generalized product of experts (gPoE) and Bayesian committee machine (BCM), it is non-trivial to determine which agent's local predictions can be used to aggregate in the cloud to ensure the convergence of the cloud-based aggregated GPR. Even so, we give the novel theoretical guarantees, i.e., the convergence of the global predictive mean and the boundedness of global variance in the attack scenario. To the best of our knowledge, this is the first paper that studies Byzantine tolerance in the context of GPR.
>
> W2: Attack is perhaps too simple.
>
> $Answer:$ Thank you so much for your comment on our attack model. By reviewing the existing state-of-the-art papers with respect to Byzantine resilience, our attacks are modeled in a standard way. Please refer to papers [A1]-[A5] I list below, which are written by researchers in security community.
>
> Basically, Byzantine attack model allows the attacker to corrupt messages in an arbitrary way, and it does not impose any restriction on attack actions. It includes other complicated attack models as special cases; e.g., attack actions are obtained via solving optimization problems or games or attack actions force aggregate gradient to be null (Please see [24]). Moreover, Byzantine attack model treats attacks as black boxes, while complicated attack models treat attacks as white boxes. It is like any bounded uncertainty and a sinusoidal uncertainty generated by an exosystem. Therefore, our attack model is strong.
>
> W3: Comparison with other methods is insufficient, for instance, (a) Gaussian process with differential privacy, https://arxiv.org/abs/2106.00474 (b) Reference [23].
>
> $Answer:$ Thank you so much for your comment on comparisons between different methods. Our paper studies Byzantine tolerance of federated learning, and it is a security problem. Paper (https://arxiv.org/abs/2106.00474) studies privacy protection to Gaussian processes via differential privacy, and it is a privacy problem. Note that information security dedicates to protecting against unauthorized access, and its goals are confidentiality, integrity, and availability (CIA triad). Privacy defines the ability to secure personally identifiable data. Since adversaries can infer private information of data owners from collected raw data, in general privacy involves cryptography to make the data confidential. Since the two papers focus on different problems, comparison between them could be unfair.
>
> On the other side, reference [23] provides several aggregation rules using local predictions. However, our current theoretical results only support the proposed aggregation rule. In particular, for different aggregation rules, e.g., gPoE and BCM, theoretical results could be completely different and they are not included to avoid confusion. Our current theoretical and experimental results have demonstrated that the proposed algorithm in our paper is robust to Byzantine attacks. Generalizing the theoretical results and comparing the performances of different aggregation rules are definitely part of our future work.
>
> W4: The following reference also discussed about the malicious protection and privacy issue of distributed GP for IoT applications, which may be mentioned in the manuscript: https://ieeexplore.ieee.org/abstract/document/9250516.
>
> $Answer:$ Thank you for the reference and we have added this paper into our reference list. It is mentioned in the first paragraph of our introduction.

---

> ### Author Response · Authors · 2022-08-02
> **Response to Reviewer dgxv (Cont'd)**
>
> Questions:
>
> Q1: Do assumptions 2 and 3 hold even for the SE kernel? This is not justified in the paper.
>
> $Answer:$ Thank you so much for the careful reading. In our revised version, we add the justification after Assumption 2.
>
> First of all, the squared-exponential kernel is given as $k(z,z_*)=\sigma_f^2\exp({-\frac{1}{2\ell^2}}||z-z_*||^2)$. Based on the distance definition $D(z,z_*)=||z-z_*||$, we have that $k(z,z_*)=\kappa(D(z,z_*))$ where $\kappa(D(z,z_*))=\sigma_f^2\exp({-\frac{1}{2\ell^2}}D(z,z_*)^2)$. Hence the decomposition property in Assumption 2 is satisfied.
>
> Second, it can be seen that $\kappa(D(z,z_*))$ is a monotonically decreasing function with regard to $D(\cdot,\cdot)$. When $D(z,z_*)=0$, we have $\kappa(0)=\sigma_f^2$.
>
> Third, please refer to equation (6.5) on page 131 of reference [A8], the function can be written as $f(x)=\sum_{i=1}^{n}\alpha_ik(z_*,z_i)$ with $z_i\in\mathcal{Z}$ and $\alpha_i\in\mathbb{R}$. Since $k(z_i,z_*) = \sigma_f^2\exp({-\frac{1}{2\ell^2}}||z_i-z_*||^2)$ is Lipschitz continuous [A8],[A9], then we can conclude that Assumption 3 holds even for the SE kernel.
>
> Q2: Are the bounds tight in Theorem 1 and 2?
>
> $Answer:$ Thank you so much for the question on the tightness of the bounds. At this moment, we do not have any claim on the tightness of the bounds in Theorem 1 and 2. There have been a very limited number of numerical algorithms whose upper bounds on convergence are proven to be tight under restrictive conditions. For example, paper [A6] derives a tight upper bound of the gradient descent method with exact line search for unconstrained optimization when the objective function is $L$-smooth and $\mu$-strongly convex. Paper [A7] derives an almost tight upper bound, by comparing the upper bound with the lower bound, for nonparametric regression using stochastic gradient descent when there is no observation noise, the estimation error diminishes almost surely, and the feature variables are uniformly
> bounded. To the best of our knowledge, we have not found any paper showing the upper bound on prediction error under Byzantine attacks is tight through mathematical proofs or simulations. Please refer to [A1]-[A5] and the references therein. In fact, the upper bound provides insights on the potential factors that affect prediction errors, e.g., the number of Byzantine agents. This helps us to analyze the effects on the prediction error convergence Byzantine agents cause. We still thank for the comment, and we will consider tighter upper bounds in future works.
>
> References
>
> [A1] P. Blanchard, E. M. El Mhamdi, R. Guerraoui, and J. Stainer, “Machine learning with adversaries:
> Byzantine tolerant gradient descent,” in Proceedings of International Conference on Neural
> Information Processing Systems, pp. 118–128, 2017.
>
> [A2] J. So, B. Güler, and A. S. Avestimehr, “Byzantine-resilient secure federated learning,” IEEE
> Journal on Selected Areas in Communications, vol. 39, no. 7, pp. 2168–2181, 2020.
>
> [A3] D. Yin, Y. Chen, R. Kannan, and P. Bartlett, “Byzantine-robust distributed learning: Towards
> optimal statistical rates,” in Proceedings of the International Conference on Machine Learning,
> pp. 5650–5659, 2018.
>
> [A4] D. Data and S. Diggavi, “Byzantine-resilient high-dimensional SGD with local iterations on
> heterogeneous data,” in Proceedings of the International Conference on Machine Learning,
> pp. 2478–2488, 2021.
>
> [A5] Y. Chen, L. Su and J. Xu, ``Distributed statistical machine learning in adversarial settings: Byzantine gradient descent,'' Proceedings of the ACM on Measurement and Analysis of Computing Systems, vol. 1, no. 2, pp. 1–25, 2017.
>
> [A6] E. De Klerk, F. Glineur, and A. B. Taylor, “On the worst-case complexity of the gradient method with exact line search for smooth
> strongly convex functions,” Optimization Letters, vol. 11, no. 7, pp. 1185–1199, 2017.
>
> [A7] R. Berthier, F. Bach, and P. Gaillard, “Tight nonparametric convergence ratesfor stochastic gradient descentunder the noiseless
> linear mode,” in Proc. Advances in Neural Information Processing Systems (NeurIPS), vol. 33, 2020, pp. 2576–2586.
>
> [A8] C. K. Williams and C. E. Rasmussen, Gaussian processes for machine learning. MIT press Cambridge, MA, 2006.
>
> [A9] A. Lederer, J. Umlauft and S. Hirche, ``Uniform error bounds for Gaussian process regression with application to safe control,'' in 3rd Conference on Neural Information Processing Systems, 2019.

---

### Meta-Review · Area_Chair_Ucog · 2022-08-23

**Recommendation:** Accept
**Confidence:** Less certain

**Metareview:**

This paper proposes a novel Byzantine-robust method for performing Gaussian process regression in a Federated learning setup. The contributions include the proposed method (novel), an accompanying theoretical analysis, and experiments.

The reviewers were generally favorable about this paper, finding that the approach is novel, and appreciating the theoretical guarantees. Several criticisms were raised in the initial reviews and these were resolved in the post-rebuttal discussion. While the paper may have some limitations, there was consensus that the paper should be accepted. When preparing a camera ready version, please take care to improve the presentation and address common concerns that came up in the reviews, such as more clearly describing the relationship to previous work, and establishing the relationship between the attack model considered in this paper and that of previous work.

**Award:**

No

---

### Decision · Program_Chairs · 2022-09-14

Accept